# Oxidative stress induces stem cell proliferation via TRPA1/RyR-mediated Ca$^{2+}$ signaling in the *Drosophila* midgut

Chiwei Xu[1], Junjie Luo[2,3], Li He[1], Craig Montell[2,3], Norbert Perrimon[1,4]*

[1]Department of Genetics, Harvard Medical School, Boston, United States; [2]Department of Molecular, Cellular and Developmental Biology, University of California Santa Barbara, Santa Barbara, United States; [3]Neuroscience Research Institute, University of California, Santa Barbara, Santa Barbara, United States; [4]Howard Hughes Medical Institute, Harvard Medical School, Boston, United States

**Abstract** Precise regulation of stem cell activity is crucial for tissue homeostasis and necessary to prevent overproliferation. In the *Drosophila* adult gut, high levels of reactive oxygen species (ROS) has been detected with different types of tissue damage, and oxidative stress has been shown to be both necessary and sufficient to trigger intestinal stem cell (ISC) proliferation. However, the connection between oxidative stress and mitogenic signals remains obscure. In a screen for genes required for ISC proliferation in response to oxidative stress, we identified two regulators of cytosolic Ca$^{2+}$ levels, transient receptor potential A1 (TRPA1) and ryanodine receptor (RyR). Characterization of TRPA1 and RyR demonstrates that Ca$^{2+}$ signaling is required for oxidative stress-induced activation of the Ras/MAPK pathway, which in turns drives ISC proliferation. Our findings provide a link between redox regulation and Ca$^{2+}$ signaling and reveal a novel mechanism by which ISCs detect stress signals.

*For correspondence: perrimon@receptor.med.harvard.edu

Competing interests: The authors declare that no competing interests exist.

## Introduction

Multipotent intestinal stem cells (ISCs) are responsible for tissue homeostasis in the adult *Drosophila* midgut (*Jiang and Edgar, 2011*; *Micchelli and Perrimon, 2006*; *Ohlstein and Spradling, 2006*). Similar to the mammalian digestive epithelium, the activity of ISC proliferation and differentiation increases upon tissue damage (*Amcheslavsky et al., 2009*). Although signaling pathways controlling ISC activity, such as Ras/MAPK (*Jiang et al., 2011*), Hippo/Yki (*Karpowicz et al., 2010*; *Shaw et al., 2010*), and JAK/Stat (*Jiang et al., 2009*), play critical roles in gut homeostasis, the mechanisms modulating these pathways under different physiological and pathological conditions are not well understood. In particular, it is unclear how ISCs sense microenvironment signals under tissue damage conditions to upregulate the activity of mitogenic pathways such as Ras/MAPK. The simplicity of the *Drosophila* midgut and the ease of performing genetic screens provide a powerful system to study how stem cells sense microenvironment cues and adjust their activities accordingly.

The concentration of reactive oxygen species (ROS) is emerging as a critical microenvironment signal that regulates stem cell activity in neuronal and glial progenitors (*Liu et al., 2009*; *Smith et al., 2000*; *Tsatmali et al., 2005*), as well as in mammalian hematopoietic stem cells (HSCs) (*Tothova et al., 2007*) and *Drosophila* hematopoietic progenitors (*Owusu-Ansah and Banerjee, 2009*). In most cases, high ROS concentration stimulates, at least in the short term, the activity of stem cells by driving them out of an otherwise quiescent status. In mammals, oxidative stress is associated with intestine damage and inflammatory bowel disease (*Rezaie et al., 2007*). In the *Drosophila* midgut, high ROS levels are produced in response to different types of tissue damage (*Ha et al.,*

*2005*; *Hochmuth et al., 2011*; *Lee et al., 2013*). Moreover, oxidants such as paraquat and hydrogen peroxide are commonly used to induce tissue damage (*Chatterjee and Ip, 2009*). Although high ROS levels in ISCs are sufficient to trigger ISC proliferation (*Hochmuth et al., 2011*), the connection between oxidative stress and mitogenic signals in ISCs remains obscure.

We performed an in vivo RNAi screen to identify novel components required for ISC proliferation in response to tissue damage. In the screen, we identified two cation channels, transient receptor potential A1 (TRPA1) and ryanodine receptor (RyR), which are required for ISC self-renewal and damage-induced proliferation. We show that these channels are required for $Ca^{2+}$ homeostasis in ISCs and mediate increases in cytosolic $Ca^{2+}$ in response to oxidative stress and ISC proliferation. Our results are consistent with a recent study (*Deng et al., 2015*) showing that high cytosolic $Ca^{2+}$ in ISCs can stimulate proliferation. Finally, we characterize the signaling events downstream of cytosolic $Ca^{2+}$ increase and demonstrate that Ras/MAPK mediates the effects of $Ca^{2+}$ on ISC proliferation.

## Results

### TRPA1 and RyR are required for stem cell damage response and self-renewal

From an RNAi screen for new transmembrane proteins required for ISCs proliferation, we identified TRPA1 and RyR as candidate hits required for ISC pool size maintenance and damage-induced ISCs proliferation (see Materials and methods). We confirmed that the lines producing a consistent phenotype (v37249, BL31504 for TRPA1; BL29445 for RyR) effectively knockdown the corresponding transcripts, while the lines that did not score as hits from our screen (BL31384, BL36780 for TRPA1; BL31540, BL31695 for RyR) cannot achieve efficient knockdown (*Figure 1—figure supplement 1C–D*).

TRPA1 is a $Ca^{2+}$-permeable cation influx channel (*Wang et al., 2013*). RyR is an endoplasmic reticulum (ER) cation channel that releases ER $Ca^{2+}$ into the cytosol (*Santulli and Marks, 2015*). RNAi-mediated knockdown of either *trpA1* or *RyR* in adult ISCs completely blocked proliferation induced by the tissue-damaging agent paraquat or bleomycin (*Figure 1A–C and A'-C', G*). Depletion of *trpA1* or *RyR* causes a gradual loss of the stem cell population, with a significant decrease in esg+ cell density undetectable until 7 days of RNAi expression (*Figure 1—figure supplement 1E*). The RNAi lines do not cause stem cell apoptosis, because cell death cannot be detected with anti-cleaved-caspase 3 staining in the midgut (*Figure 1—figure supplement 2A–C*), and the anti-apoptotic gene *p35* (*Hay et al., 1994*) cannot rescue the mitosis defect caused by *trpA1* RNAi or *RyR* RNAi (*Figure 1—figure supplement 2D–I,D'–I'*). In contrast, mitosis activity in stem cells expressing *trpA1* RNAi or *RyR* RNAi can be regained by forced cytosolic calcium influx with RNAi against *SERCA* (*Figure 1D–G*), an ER $Ca^{2+}$ ATPase whose inhibition triggers Stim/Orai channel (SOC)-mediated $Ca^{2+}$ entry into the cytosol (*Clapham, 2007*). Altogether, these results place cytosolic calcium signaling genetically downstream of TRPA1 and RyR during ISC proliferation. Consistent with the results obtained with RNAi, CRISPR/Cas9-mediated knockout of *trpA1* or *RyR* in adult ISCs suppressed damage-induced proliferation (*Figure 1H*). In addition, at the organ level, inhibition of ISC proliferation by *trpA1* RNAi is associated with a shortening in midgut length (*Figure 1I*). Further, while wild-type midguts maintain their proper length after feeding paraquat or bleomycin for 2 days, midguts with *trpA1* RNAi expression in the ISCs exhibit accelerated shortening following tissue damage. This damage-induced midgut shortening phenotype is reminiscent to the gut phenotype observed after depletion of the esg+ cell population when the cell death gene *reaper* (*rpr*) is expressed in ISCs for 4 days (*Figure 1—figure supplement 3*), suggesting the importance of ISC activity for maintaining tissue size.

To determine whether TRPA1 and RyR act in a cell-autonomous manner, we generated GFP+ clones originating from ISCs that are deficient for either *trpA1* or *RyR* using the MARCM method (*Wu and Luo, 2006*). MARCM clones generated from ISCs expressing *trpA1* RNAi (*Figure 2A'*), or homozygous for the loss-of-function allele *trpA1[1]* (*Kwon et al., 2008*) (*Figure 2B'*), or *RyR[k04913]* (*Sullivan et al., 2000*) (*Figure 2C'*) had a smaller average size compared to wild-type clones 10 days after clonal induction (*Figure 2A–C*, *trpA1* RNAi and *RyR[k04913]* clone sizes are quantified in *Figure 2D*), suggesting that TRPA1 and RyR are required autonomously for clonal expansion.

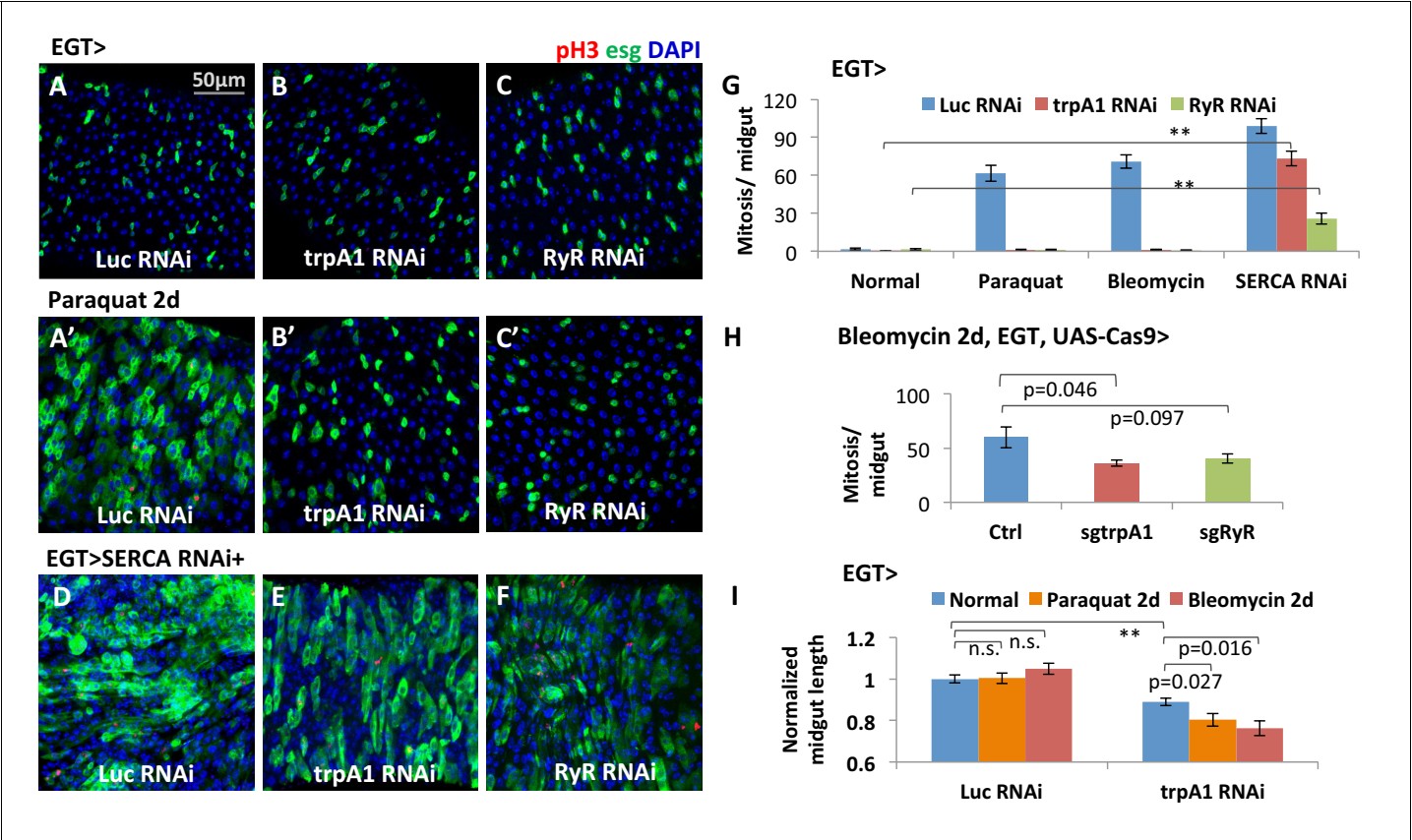

**Figure 1.** The calcium channels TRPA1 and RyR are required for damage-induced ISC proliferation. (A–C) Midguts overexpressing *Luc* RNAi, *trpA1* RNAi or *RyR* RNAi in ISCs using EGT (esgGal4 UAS-GFP tubGal80[ts]) for 6d, were fed the oxidant agent paraquat for the last 2d (A'–C') or co-expressed with *SERCA* RNAi (D–F). Midguts are stained for the mitosis marker phosphohistone H3 (pH3). *Luc* RNAi is used as control. [ts] stands for temperature sensitive tubGal80ts that allows for temporal control of genetic manipulation. GFP is used to report the expression pattern of esgGal4. (G) Quantification of mitosis (pH3+ cell number) of midguts expressing *Luc* RNAi, *trpA1* RNAi or *RyR* RNAi in ISCs under normal condition, 2 mM paraquat feeding, 25 μg/ml bleomycin feeding, or co-expression with *SERCA* RNAi. N > 5 midguts are quantified per genotype per treatment. Data are represented as mean ± SEM. Double asterisks indicate a p value of less than 0.01. (H) Mitosis quantification of midguts with constitutive expression of sgRNA targeting *trpA1* or *RyR*, and targeted Cas9 expression in ISCs for 7d (bleomycin feeding for the last 2d). Flies with the same genetic background but only empty insertional landing site are used as the control for sgRNA. Data are represented as mean ± SEM. Note that we generally observed that CRISPR/Cas9-mediated knockout in vivo is not working as efficiently as RNAi, probably because not all DNA damages by CRISPR/Cas9 result in frame-shift mutations. N > 6 midguts are analyzed per genotype. Data are represented as mean ± SEM. (I) Length quantification of midguts expressing *Luc* RNAi (control) or *trpA1* RNAi for 7d, with the last 2d feeding on normal food, paraquat, or bleomycin. The average midgut length of control flies feeding on normal food is used for normalization. N > 5 midguts are analyzed per genotype per treatment. Data are represented as mean ± SEM.

The following source data and figure supplements are available for figure 1:

**Source data 1.** Complete results for *Figure 1G–I*, *Figure 1—figure supplement 1E*, *Figure 1—figure supplement 3A–B*.

**Figure supplement 1.** Validation of knockdown efficiency for *trpA1* RNAi and *RyR* RNAi lines.

**Figure supplement 2.** *trpA1* RNAi and *RyR* RNAi do not cause ISC apoptosis.

**Figure supplement 3.** ISC depletion results in midgut shortening.

Although smaller than wild-type clones, MARCM clones expressing *trpA1* RNAi or losing *RyR* can survive and differentiate into Pdm1+ enterocytes (ECs) (*Figure 2—figure supplement 1A–D*). We also performed lineage tracing with the EGT F/O system (*Jiang and Edgar, 2009*), which relies on induced expression of Flp in the stem cells to excise transcriptional STOP cassettes flanking act::

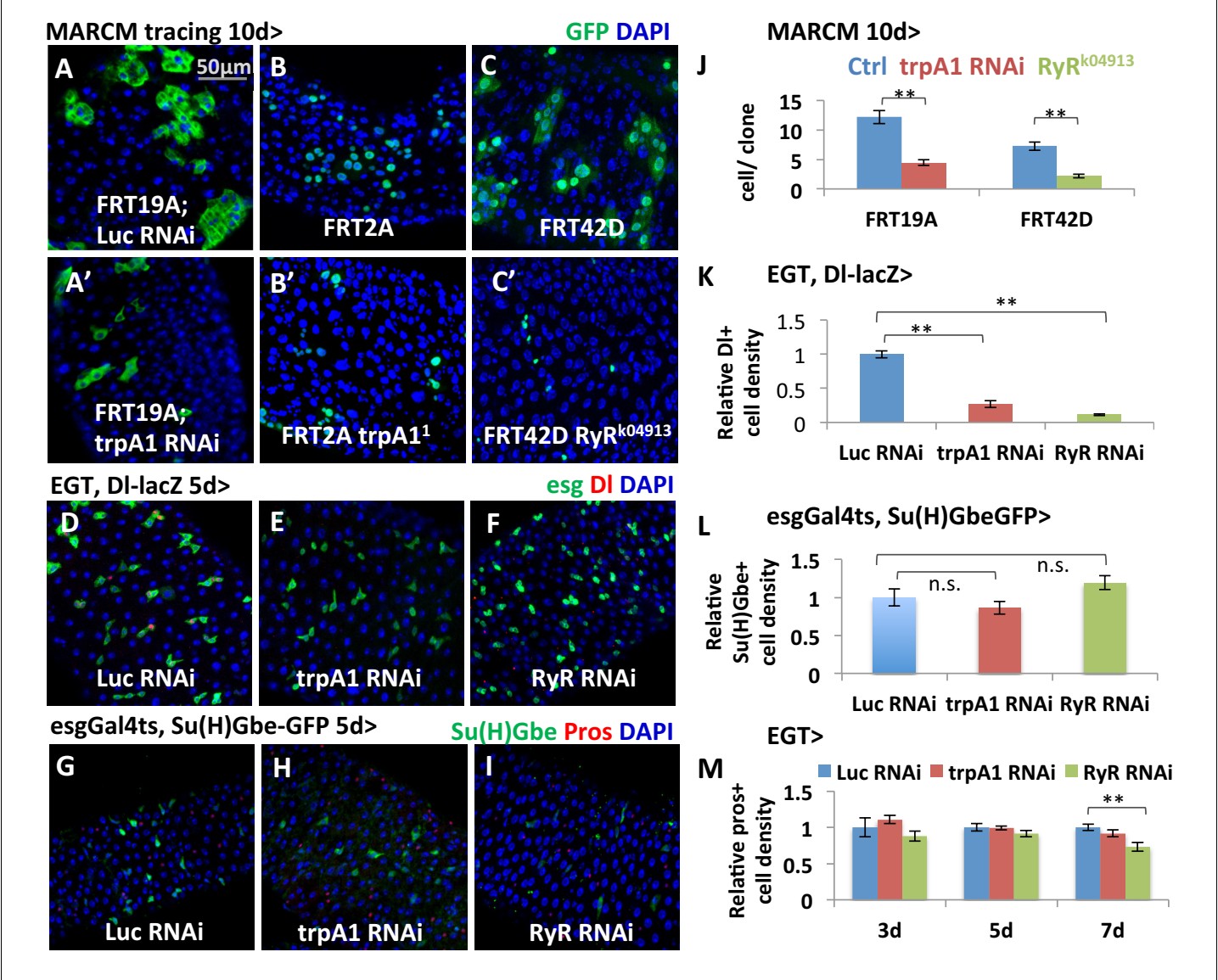

**Figure 2.** TRPA1 and RyR are required for ISC self-renewal but not differentiation. (A–C, A'–C') MARCM clones deficient for TRPA1 or RyR are analyzed along with their corresponding control genotypes (A for A', B for B', C for C') 10d after clone induction. Cell numbers per clone for MARCM clones expressing *trpA1* RNAi, or lacking both wild-type *RyR* alleles, are quantified in (J). Randomly picked N = 35 clones from the posterior regions of five guts per genotype are analyzed. Data are represented as mean ± SEM. (D–F) ISC marker Dl-lacZ stainings of midguts expressing *Luc* RNAi, *trpA1* RNAi or *RyR* RNAi for 5d in ISCs. The relative Dl+ cell density in the posterior midgut region is quantified in (K). Dl+ cell number is divided by the imaged area size for density calculation. N > 5 midguts per genotype are analyzed. Data are represented as mean ± SEM. (G–I) Immunostaining of midguts expressing *Luc* RNAi, *trpA1* RNAi or *RyR* RNAi for 5d in ISCs for Notch activity reporter Su(H)GbeGFP and EE marker Prospero (pros). The relative Su(H)Gbe+ cell density in the posterior midgut region is quantified in (L). Su(H)Gbe+ cell number is divided by the imaged area size for density calculation. N > 4 midguts per genotype are analyzed. Data are represented as mean ± SEM. (M) Quantificatioin of EE density in the posterior midgut region for midguts expressing *Luc* RNAi, *trpA1* RNAi or *RyR* RNAi in ISCs for 3d, 5d, and 7d. For each time point, EE density is normalized to the average value of midguts expressing *Luc* RNAi. N > 4 midguts per genotype are analyzed. Data are represented as mean ± SEM.

The following source data and figure supplement are available for figure 2:

**Source data 1.** Complete results for *Figure 2J–M*.
**Figure supplement 1.** Lineage-tracing experiments provide evidence that TRPA1 or RyR-deficient ISCs can differentiate.

Gal4, and found that stem cells expressing *trpA1* RNAi can still differentiate into mature Pros+ enteroendocrine cells (EEs) or Pdm1+ ECs (*Figure 2—figure supplement 1E–H*). Furthermore, while Dl+ cells (*Figure 2D–F and L*) are dramatically decreased by *trpA1* RNAi or *RyR* RNAi expression for 5 days, there was no significant change in the density of Su(H)Gbe+ or Pros+ cells (*Figure 2G–I and M*). The Notch pathway reporter Su(H)GbeGFP is indicative of ISC differentiation activity because Notch signaling is high when ISCs differentiate, especially for the EC lineage (*Micchelli and Perrimon, 2006*; *Ohlstein and Spradling, 2007*). Altogether, our data suggest that *trpA1* RNAi or *RyR* RNAi mainly affects ISC self-renewal and proliferation in the adult gut, rather than ISC differentiation or cell death.

## TRPA1 and RyR expression in the midgut

The *trpA1* gene is expressed as multiple mRNAs (*Figure 1—figure supplement 1A*), four of which have been characterized previously (*Zhong et al., 2012*). Compared to the TRPA1-A and -B isoforms, expressed in the fly head, TRPA1-C and -D isoforms are expressed in the midgut (*Figure 1—figure supplement 1B*) and have different N-terminal residues that might confer sensitivity to different stimuli. Based on anti-TRPA1 staining, TRPA1 appears to be expressed in stem cells (*Figure 3A–B*). Using a Gal4 fused to the C terminal of *trpA1* transcripts via CRISPR/Cas9-mediated knock-in at the endogenous locus (*Figure 3—figure supplement 1*), we could also detect *trpA1* expression in stem cells in the anterior region of the posterior midgut (*Figure 3C*). Consistently, stem cell depletion reduces *trpA1* total mRNA expression in the midgut by more than 50% (*Figure 3—figure supplement 2A*). Because the signal-to-noise ratio for anti-TRPA1 staining is low, and *trpA1*Gal4[CP2A] might not fully recapitulate the endogenous expression pattern, we cannot exclude the possibility that *trpA1* is also expressed in other cell types. In a recent study, for example, TRPA1-C has been identified to regulate defecation of food-borne pathogens in the EEs (*Du et al., 2016*). However, TRPA1 is mainly required in ISCs for proliferation, because knockdown of *trpA1* in self-renewing ISCs, rather than in differentiating ISCs (Su(H)Gbe+ cells, also called enteroblasts (EBs)) or visceral muscles, blocks damage-induced proliferation (*Figure 3—figure supplement 2B*). Further, an additional piece of evidence that TRPA1 protein functions in the stem cells comes from the ex vivo calcium-imaging experiments with GCaMP6f reporter, for which the intensity of green fluorescence indicates cytosolic calcium levels (*Chen et al., 2013*). While the TRPA1 agonist AITC (allyl isothiocyanate (*Bautista et al., 2005*) can induce calcium influx in ISCs carrying a wild-type allele of *trpA1*, such response is abolished in flies losing both alleles of wild-type *trpA1* (*Figure 3E–F*).

RyR appears to be ubiquitously expressed at low levels in the midgut epithelium, based on the expression pattern of *RyRGal4[R14G09]* (*Figure 3D*), where Gal4 is under the control of 2.92 kb putative enhancer region sequence of *RyR* (*Pfeiffer et al., 2008*).

## TRPA1-D in the midgut senses the oxidative stress associated with tissue damage

Oxidative stress has previously been observed in tissue damage conditions involving JNK activation (*Hochmuth et al., 2011*) and pathogen infection (*Ha et al., 2005*; *Lee et al., 2013*). The source of oxidative stress is generally attributed to an imbalance in the formation of ROS and impaired antioxidant machinery in unhealthy tissues (*Hochmuth et al., 2011*; *Lih-Brody et al., 1996*). Staining with dihydro-ethidium (DHE) (*Owusu-Ansah et al., 2008*) confirmed that ROS levels in the midgut increase following tissue damage induced by feeding flies with paraquat (*Figure 4—figure supplement 1B*), a widely used oxidant that interferes with electron transfer to catalyze formation of superoxide-free radicals (*Bus and Gibson, 1984*), or bleomycin (*Figure 4—figure supplement 1C*), a chemotherapy drug that cleaves DNA via an unresolved mechanism potentially related to ROS production (*Hecht, 2000*). In addition, induction of midgut stress by JNK pathway activation or massive EC apoptosis by *rpr* expression can also lead to increased ROS and ISC expansion (*Figure 4—figure supplement 1F–J*).

Both TRPA1 and RyR are associated with redox sensing (*Guntur et al., 2015*; *Hamilton and Reid, 2000*; *Sun et al., 2011*). Guntur et al. found that TRPA1-C and TRPA1-D isoforms are $H_2O_2$-sensitive and that their endogenous expressions in the ring gland, or ectopic expression in other tissues, confer UV light sensitivity via sensing UV light-induced $H_2O_2$ production. To directly measure whether gut-specific TRPA1 isoforms can respond to paraquat, we used two-electrode voltage clamp

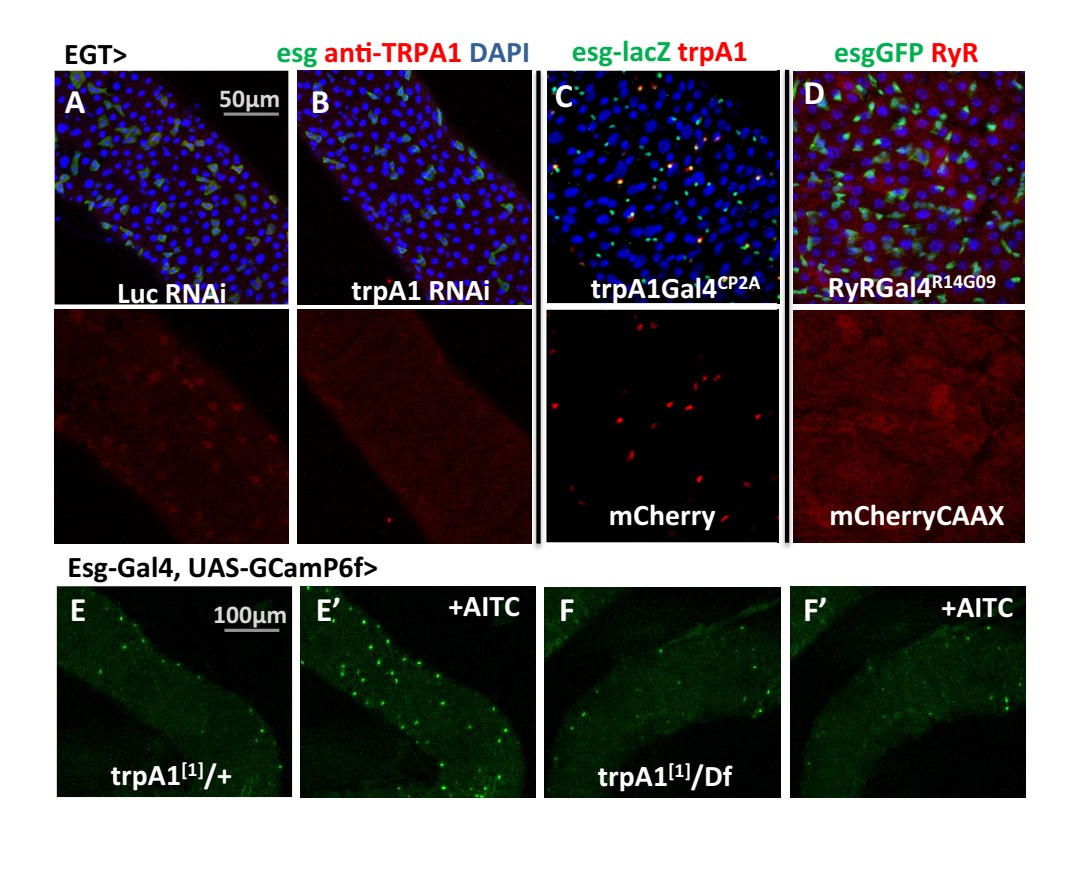

**Figure 3.** *trpA1* and *RyR* expression in the midgut. (A–B) Anti-TRPA1 immunostaining of midguts expressing *Luc* RNAi or *trpA1* RNAi in ISCs. The channel of anti-TRPA1 signal is shown below the merged image. Note that anti-TRPA1 staining has high background signal. While we could detect TRPA1 expression in the ISCs, we could not exclude the possibility that it is also expressed in other cell types. (C) *trpA1*Gal4[CP2A]-driven expression of mCherry is co-stained with ISC marker esglacZ. (D) *RyR*Gal4[R14G09] (enhancer region of RyR locus) driven expression of membrane-localized CAAX-mCherry is co-stained with ISC marker esgGFP. (E–F) Imaging midguts missing one or both alleles of wild-type *trpA1* with GCaMP6f expression in ISCs. The same guts are imaged again after exposure to 0.015% AITC in the imaging buffer for 5 min (E′–F′). The images are acquired on a Keyence microscope.

The following source data and figure supplements are available for figure 3:

**Source data 1.** Complete results for *Figure 3—figure supplement 2B*.

**Figure supplement 1.** The knock-in design of *trpA1*Gal4[CP2A].

**Figure supplement 2.** Additional evidence for *trpA1* expression and function in ISCs.

recordings to measure the channel activities of TRPA1-C and -D (*Wagner et al., 2000*). In this experiment, we expressed similar amount of mRNA for TRPA1-C or -D isoform in the *Xenopus* oocytes, clamped the membrane potential at −80 mV, and measured the current flowing through the ion channels in the oocyte membrane. The current value is proportional to the opening probability of the ion channels. We added paraquat and AITC (used as the positive control) into the bath solution of oocytes sequentially to trigger the opening of the TRPA1-C or -D channel and compare their strength and dynamics. TRPA1-C barely displayed any response to paraquat (*Figure 4A*), whereas expression of TRPA1-D induced a slow increase in current (*Figure 4B*). Consistently, overexpression of TRPA1-D, but not TRPA1-C, stimulated ISCs expressing *trpA1* RNAi to proliferate in response to paraquat (*Figure 3C–F*).

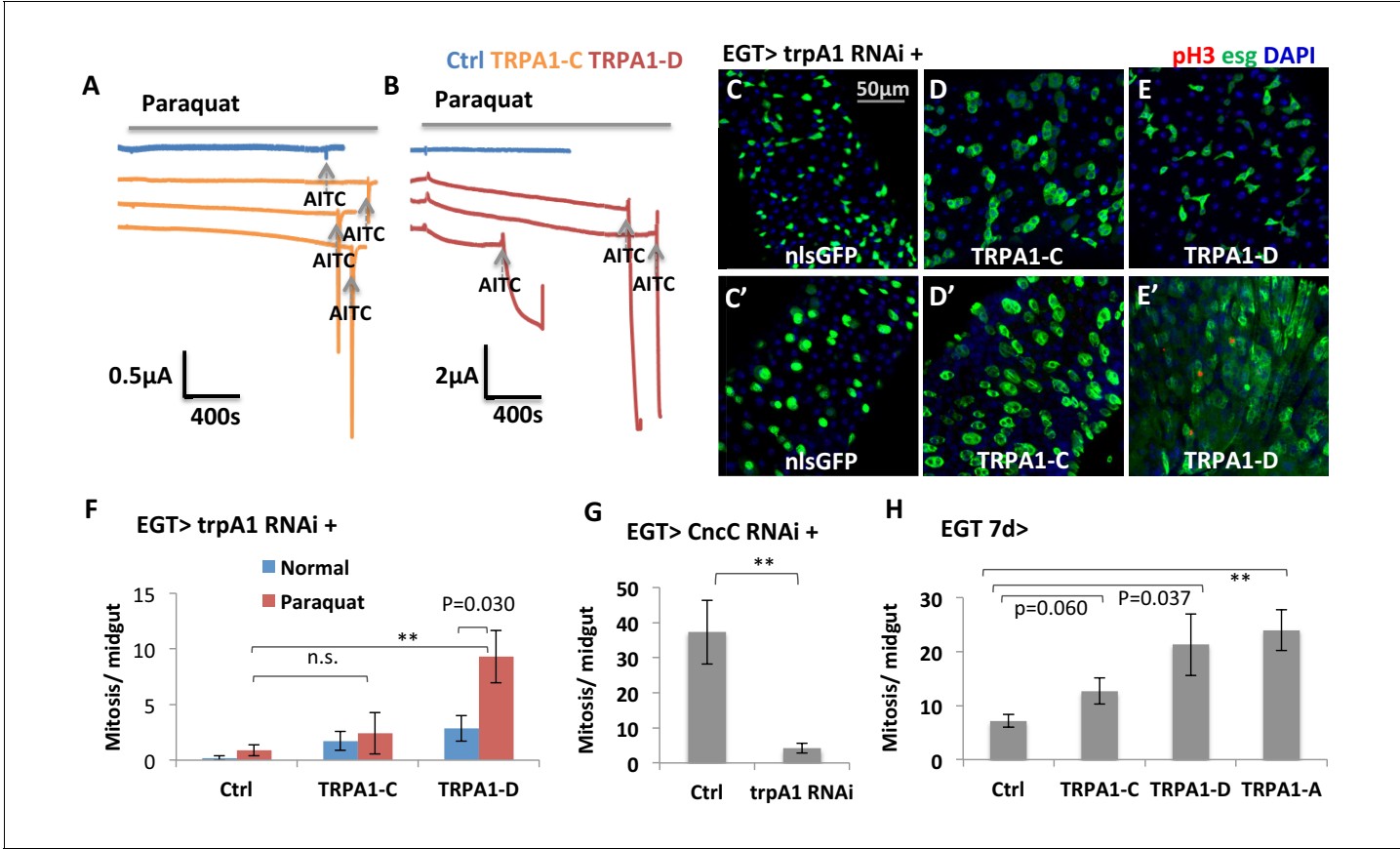

**Figure 4.** TRPA1 can respond to oxidant agent and its activation stimulates ISC activity. (**A–B**) Oocyte clamp measurement of TRPA1-C or TRPA1-D channel activities in response to paraquat. The same amounts of mRNAs for TRPA1-C or TRPA1-D were injected into Xenopus oocytes. 4 mM paraquat is present in the buffer during the period marked with the gray line on the top. Near the end of the experiment, the TRPA1 agonist AITC was added as a positive control (indicated with arrowhead). 2 mM and 10 mM paraquat were also tested and gave consistent results. (**C–E**) Midguts, expressing *trpA1* RNAi alone, or together with TRPA1-C or TRPA1-D in the ISCs for 7d, with the last 2d feeding paraquat (**C'–E'**), are stained for the mitosis marker pH3. Nucleus-localized GFP (nlsGFP) is used as the control for transgenic expression. We have also tested UAS-ultraGFP (*Yang and Tower, 2009*) that consists of multiple copies of UAS-2xEGFP as control and obtained the same results as UAS-nlsGFP with regard to stem cell activity (except that ultraGFP reporter is much brighter than nlsGFP). (**F**) Mitosis quantification of midguts expressing *trpA1* RNAi together with TRPA1-C or TRPA1-D under normal and paraquat-feeding conditions. N > 4 midguts per genotype per treatment are analyzed for quantification. UAS-ultraGFP is used as the control for transgenic expression. Data are represented as mean ± SEM. Note that TRPA1-C or TRPA1-D are still targeted by *trpA1* RNAi, which might limit their capacity to rescue *trpA1* RNAi. (**G**) Mitosis quantification of midguts expressing *CncC* RNAi alone, or together with *trpA1* RNAi in the ISCs for 7d. N > 7 midguts are analyzed for each genotype. Data are represented as mean ± SEM. (**J**) Mitosis quantification of midguts overexpressing TRPA1-C, TRPA1-D, or TRPA1-A in ISCs for 7d. N > 5 midguts per genotype are analyzed. Data are represented as mean ± SEM.

The following source data and figure supplement are available for figure 4:

**Source data 1.** Complete results for *Figure 4F–H*.

**Figure supplement 1.** Elevated ROS is a common stress signal in various midgut damage conditions.

Previously, it has been reported that knockdown of antioxidant protein CncC in ISCs can induce mitosis (*Hochmuth et al., 2011*). Our observation that *trpA1* RNAi suppresses the hyper-proliferation status induced by CncC RNAi (*Figure 4G*) places TRPA1 genetically downstream of the pathway controlling ROS for ISC proliferation. Further, overexpression of TRPA1-D or conventional TRPA1-A isoform (active at 29°C) in stem cells can induce mitosis significantly, suggesting that increased TRPA1-calcium signaling is sufficient to drive ISC proliferation (*Figure 4H*).

## TRPA1 and RyR are required for normal and ROS-induced cytosolic Ca$^{2+}$ levels in ISCs

Interestingly, TRPA1 and RyR are not critically required in the midgut epithelium for development, because flies expressing *trpA1* RNAi or *RyR* RNAi with esgGal4 throughout development [that is, in adult midgut progenitors (AMPs) during larval stages and in adults ISCs, see (*Jiang and Edgar, 2009*) can survive to adulthood without gross defects when they are fed with normal food]. For young adult flies of these genotypes, we used the ultrasensitive GCaMP6s fluorescent reporter (*Chen et al., 2013*) to examine cytosolic Ca$^{2+}$ concentrations in their ISCs. While the number of GFP + (i.e. high cytosolic Ca$^{2+}$ concentration) ISCs in wild-type midguts increases significantly 10 min after adding paraquat to a final concentration of 4 mM (*Figure 5A, A' and D*), expression of *trpA1* RNAi or *RyR* RNAi in ISCs (driven by esgGal4 throughout development) not only reduced the basal levels of cytosolic Ca$^{2+}$ in stem cells (*Figure 5A–C*) but also inhibited the ROS-mediated increase in cytosolic Ca$^{2+}$ (*Figure 5A'–C'*). The observed reduction of high Ca$^{2+}$ ISCs is not due to decrease in the total number of ISCs, as total ISC number or density in young adult flies is not yet significantly affected by *trpA1* RNAi or *RyR* RNAi (*Figure 5—figure supplement 1*).

Defining the average number of high Ca$^{2+}$ stem cells before drug treatment as a baseline, we used the confocal time-lapse GCaMP6s imaging (*Videos 1–6*) and analysis to identify the kinetics of high Ca$^{2+}$ in stem cells. Consistent with the TRPA1 channel kinetics as measured in the voltage clamp experiments in oocytes (*Figure 4A–B*), treatment with 4 mM paraquat caused a slow and continuous increase in the average number of high Ca$^{2+}$ ISCs (*Figure 5E*), whereas the TRPA1 agonist AITC triggered a sharp increase followed by a quick drop in the number of high Ca$^{2+}$ ISCs (*Figure 5F*). Finally, responses to both paraquat and AITC were dependent on TRPA1 and RyR (*Figure 5E–F*), whereas treatment with the SERCA inhibitor thapsigargin, as a positive control, led to increased Ca$^{2+}$ in ISCs expressing *trpA1* RNAi or *RyR* RNAi (*Figure 5G*).

Using GCaMP5G as the Ca$^{2+}$ reporter and tdTomato as an internal control, we found that *trpA1* RNAi blocked the increase of cytosolic Ca$^{2+}$ following treatment with 2 mM paraquat (*Figure 5—figure supplement 2A,A', B,B'*). Similar results were observed with another reporter GCaMP6f (data not shown).

In addition, we used TRIC (transcriptional reporter of intracellular Ca$^{2+}$) (*Gao et al., 2015*) to examine cytosolic Ca$^{2+}$ concentration in ISCs in vivo on fixed midguts. With TRIC lexGFP and a conventional UAS-RFP reporter, we found that the percentage of high Ca$^{2+}$ ISCs among all stem cells increased from 23% to 63% (*Figure 5—figure supplement 2C,C', F*) when flies were fed 2 mM paraquat for 1 day. In adult midguts expressing *trpA1* RNAi or *RyR* RNAi in adult ISCs for 6 days, high Ca$^{2+}$ ISCs showed little, if any, increase following paraquat feeding (*Figure 5—figure supplement 2D,D', E,E', F*). Unlike GCamP reporters, a reduction in basal levels of high Ca$^{2+}$ ISCs by *trpA1* RNAi or *RyR* RNAi was not observed with TRIC reporter, which might be due to differences in the stability and detection threshold of reporters as well as the time frame of knockdown for the two sets of experiments.

## Cytosolic Ca$^{2+}$ induces Ras/MAPK activity

The biological functions of TRPA1, RyR, and cytosolic Ca$^{2+}$ signaling have been extensively studied in neurons, where it has been established that membrane depolarization and Ca$^{2+}$ influx can activate Ras/MAPK (*Randlett et al., 2015a*; *Rosen et al., 1994*). Thus, we next asked whether high Ca$^{2+}$ in ISCs similarly activates Ras/MAPK. To do this, we stained midguts with antibodies against diphospho-extracellular signal-regulated kinase (dpErk), which is a measure of *Drosophila* Ras/MAPK activity (*Gabay et al., 1997*). Adult midguts expressing *SERCA* RNAi in ISCs for 2 days, which at that stage do not present signs of ISCs expansion, showed induced MAPK activity (*Figure 6A–B*), and expression of *trpA1* RNAi (*Figure 6C*) or *RyR* RNAi (*Figure 6D*) decreased the levels of MAPK activity. Consistent with our observation that ISC Ca$^{2+}$ induction by SERCA inhibition does not depend on *trpA1* or *RyR* (*Figure 5G*), *SERCA* RNAi could override *trpA1* RNAi (*Figure 6G*) or *RyR* RNAi (*Figure 6H*) to activate MAPK. To test whether the increased dpErk signal we observed with *SERCA* RNAi merely reflects an hyper-proliferative status for ISCs, we combined *SERCA* RNAi with knockdown of *Yorkie* (*Yki*), the Hippo pathway transcriptional factor required for damage-induced ISC proliferation (*Karpowicz et al., 2010*; *Ren et al., 2010*; *Shaw et al., 2010*; *Staley and Irvine, 2010*). Although *Yki* RNAi could inhibit ISC proliferation caused by *SERCA* RNAi (*Figure 7—figure*

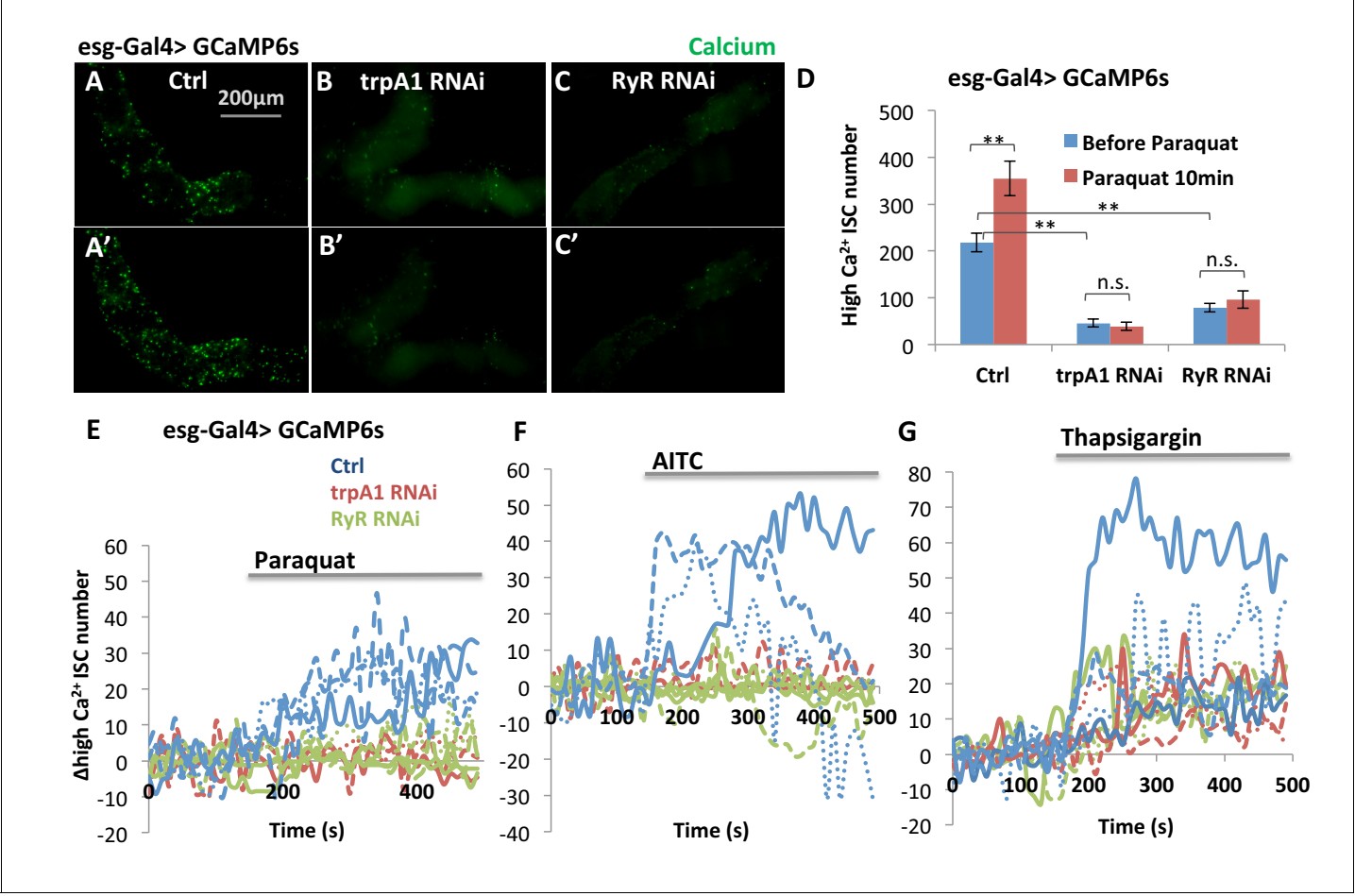

**Figure 5.** TRPA1 and RyR are required for ROS-mediated $Ca^{2+}$ increases in ISCs. (A–C) Imaging of midguts expressing the GCaMP6s reporter alone or together with *trpA1* RNAi or *RyR* RNAi in ISCs. The same guts are imaged again after exposure to 4 mM paraquat in the imaging buffer for 10 min (A'–C'). The images are acquired on a Keyence microscope. (D) Quantification of high $Ca^{2+}$ (GFP+) stem cells within the posterior midgut region. N > 5 midguts are analyzed for each genotype. Data are represented as mean ± SEM. (E–G) Numeric kinetics of high $Ca^{2+}$ stem cells in midguts expressing *trpA1* RNAi or *RyR* RNAi in ISCs. Wild-type midguts expressing GCaMP6s alone serve as control. Time-lapse confocal imaging is used for the analysis (see Supplementary Movies). The average number of high $Ca^{2+}$ stem cells before drug treatment is set as the basal level of 0 for individual imaging experiments. Final concentration of 4 mM paraquat (E), 0.03% AITC (F), 4 µM thapsigargin (G) are added at 150 s. At least three replicates of each genotypes/treatments are shown in the figure, with different colors labeling different genotypes.

The following source data and figure supplements are available for figure 5:

**Source data 1.** Results for *Figure 5D*, *Figure 5—figure supplement 1D–E*, *Figure 5—figure supplement 2F*.

**Figure supplement 1.** The total number of ISCs expressing esgGal4>GCamP6s reporter is not significantly reduced by *trpA1* RNAi or *RyR* RNAi.

**Figure supplement 2.** Additional reporters showing that TRPA1 and RyR are required for ROS-mediated $Ca^{2+}$ increases in ISCs.

supplement 1F–G), it could not block MAPK activity (*Figure 6F*). On the other hand, *Ras1* (*Drosophila Ras*) RNAi could block *SERCA* RNAi-induced MAPK activity (*Figure 6E*). Altogether, our data suggest that Ras/MAPK activity is likely the direct target of high cytosolic $Ca^{2+}$ in ISCs, rather than the consequence of cross-activation by other proliferation-related pathways such as Yki.

To further examine the relationship between $Ca^{2+}$ levels and MAPK activity, we performed dpErk staining of midguts expressing the TRIC reporter in ISCs. Paraquat feeding for 1 day led to an increase in the number of cells with high dpErk signals, a majority of which are high $Ca^{2+}$ ISCs (*Figure 6I,I'*). Moreover, paraquat induction of MAPK activity appears to require high $Ca^{2+}$ in ISCs,

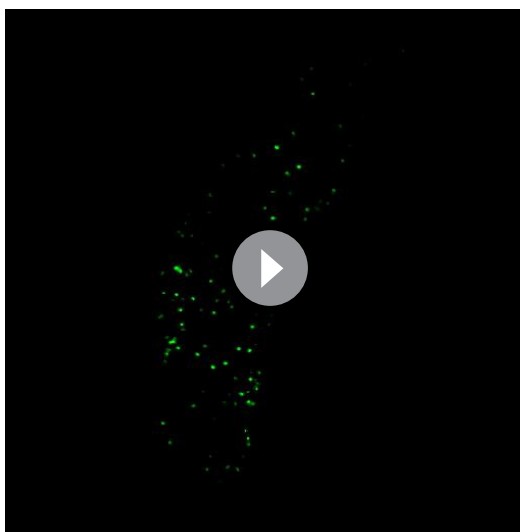

**Video 1.** Calcium imaging of ISCs in response to paraquat. Midguts expressing GCaMP6s reporter in ISCs are dissected and imaged in adult hemolymph-like (AHL) buffer. Z-stack images were acquired with 10 s interval. A maximal intensity z-projection was shown in the movie. A final concentration of 4 mM oxidant agent paraquat was added at 150 s (the 15th frame of the movie).

because such response could be inhibited by *trpA1* RNAi expression in ISCs (*Figure 6J,J'*). It should be noted that the best timing to detect autonomous activation of dpErk by high $Ca^{2+}$ in the ISCs is before they enter the hyper-proliferative stage. Once ISCs start to expand massively following prolonged high $Ca^{2+}$ induction, pErk returns to normal levels in many ISCs and is non-autonomously induced in some ECs, resulting in a diffusive activation pattern (*Figure 6—figure supplement 1A–C*).

In neurons, a common mechanism by which $Ca^{2+}$ regulates Ras/MAPK is through $Ca^{2+}$-sensitive Src kinases (*Cullen and Lockyer, 2002*), in a process faster than any transcriptional response (*Randlett et al., 2015b*; *Rosen et al., 1994*). Interestingly, overexpression of Src (Src42A or Src64B in *Drosophila*) in ISCs triggers ectopic MAPK activation and proliferation (*Figure 6—figure supplement 2A–C*), even in the presence of *trpA1* RNAi (*Figure 6—figure supplement 2E–H*). Moreover, we could observe significant inhibition of *SERCA* RNAi-induced ISC proliferation by *Src64B* RNAi (*Figure 6—figure supplement 2I*) despite the likelihood of functional redundancy between Src64B and Src42A (*Ma et al., 2013*; *Tateno et al., 2000*). These data suggest that Src

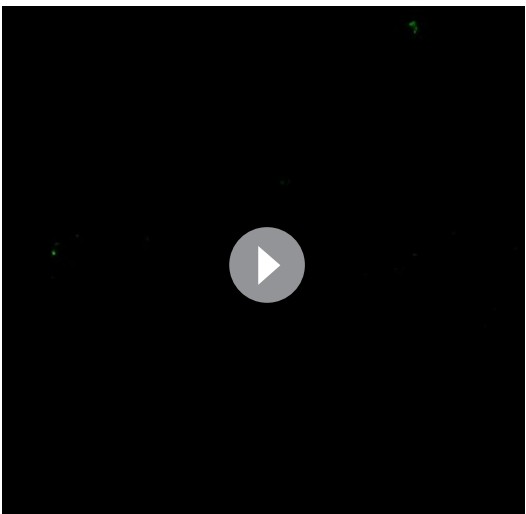

**Video 2.** Calcium imaging of ISCs expressing *trpA1* RNAi in response to paraquat. Midguts expressing GCaMP6s reporter together with *trpA1* RNAi in ISCs were dissected and imaged in AHL buffer. Z-stack images were acquired with 10 s interval. A maximal intensity z-projection was shown in the movie. A final concentration of 4 mM oxidant agent paraquat was added at 150 s (the 15th frame of the movie).

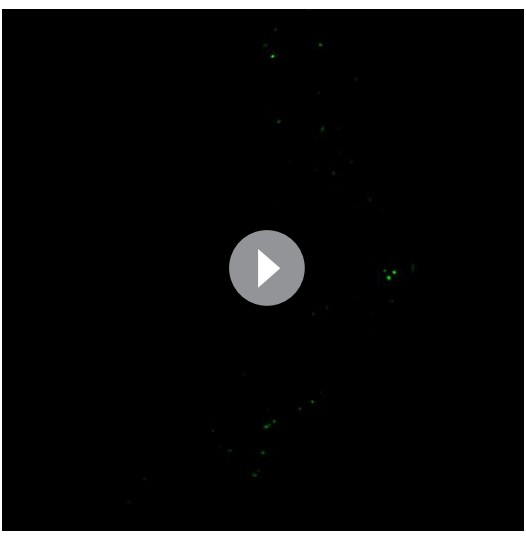

**Video 3.** Calcium imaging of ISCs expressing *RyR* RNAi in response to paraquat. Midguts expressing GCaMP6s reporter together with *RyR* RNAi in ISCs were dissected and imaged in AHL buffer. Z-stack images were acquired with 10 s interval. A maximal intensity z-projection was shown in the movie. A final concentration of 4 mM oxidant agent paraquat was added at 150 s (the 15th frame of the movie).

might mediate the activation of Ras/MAPK by cytosolic Ca$^{2+}$ in ISCs.

## Ras/MAPK activity is the major effector of cytosolic calcium for ISC proliferation

As a universal intracellular signal, increased Ca$^{2+}$ might trigger a wide range of responses. Consistent with previous results (*Deng et al., 2015*; *Jiang et al., 2011*), ISCs expressing the active forms of many Ca$^{2+}$ effector proteins, such as Ras1, Raf, phosphatase Calcineurin A1 (CanA1), CREB-regulated transcriptional co-activator (CRTC), and cyclic-AMP response element binding protein B (CrebB), displayed an hyper-proliferation phenotype (*Figure 7A–E*). To determine which Ca$^{2+}$ effector proteins are sufficient for Ca$^{2+}$-induced ISC proliferation, we expressed *trpA1* RNAi in ISCs together with each putative Ca$^{2+}$ effector protein. Strikingly, MAPK activation via expression of active Ras1 (Ras1$^A$), constitutively active Raf (Raf$^{gof}$), or secreted form of *spi* is epistatic to *trpA1* RNAi for ISCs proliferation (*Figure 7A–B and A'–B', 7F*). Consistently, MAPK activation is epistatic to *RyR* RNAi (*Figure 7—figure supplement 3*). By contrast, *trpA1* RNAi effectively blocked the mitogenic activity of constitutively active CanA1 (CanA1$^{ca}$) (*Dijkers and O'Farrell, 2007*), CRTC (*Kim et al., 2016*), or active CrebB (CrebB$^{act}$) (*Ganguly-Fitzgerald et al., 2006*) (*Figure 7C–E and C'–E', F*). Furthermore, while *Ras1* RNAi could suppress *SERCA* RNAi-induced ISC proliferation, neither *CanA1* RNAi nor *CrebB* RNAi could (*Figure 7—figure supplement 1C–G*), and *CRTC* RNAi could only confer moderate suppression of ISC hyper-proliferation (*Figure 7—figure supplement 1E–F*). Results of these genetic epistasis analyses, together with examination of changes in pErk signal intensity (*Figure 6*), indicate that MAPK activity is both necessary and sufficient for high Ca$^{2+}$-triggered ISC proliferation.

Previous studies have documented that receptor tyrosine kinase (RTK) EGFR is required for MAPK activity and ISC proliferation (*Jiang et al., 2011*). We found that *EGFR* RNAi completely inhibited ISCs proliferation caused by *SERCA* RNAi (*Figure 7—figure supplement 1B,F*), indicating that MAPK activation by Ca$^{2+}$ requires EGFR signaling. Interestingly, as transcriptional regulation of ligands and receptors by their own pathways is commonly observed (see for example *Ammeux et al., 2016*), we examined by RT-qPCR the expression of EGFR ligands, as well as other RTK ligands, in midguts with cytosolic Ca$^{2+}$ levels altered in ISCs. Interestingly, transcription of the EGFR ligand *spi* and

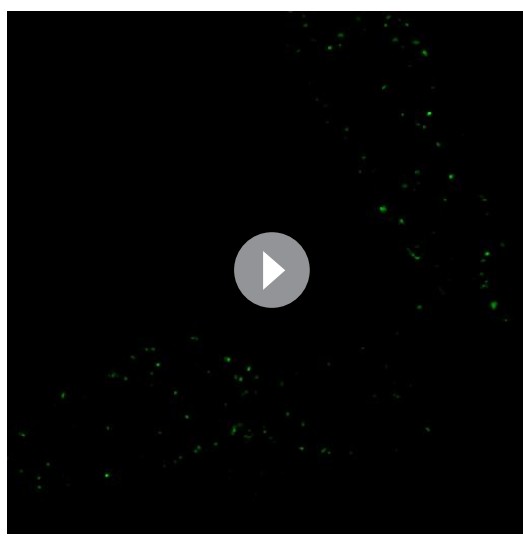

**Video 4.** Calcium imaging of ISCs in response to AITC. Midguts expressing GCaMP6s reporter in ISCs are dissected and imaged in AHL buffer. Z-stack images were acquired with 10 s interval. A maximal intensity z-projection was shown in the movie. A final concentration of 0.03% TRPA1 agonist AITC was added at 150 s (the 15th frame of the movie).

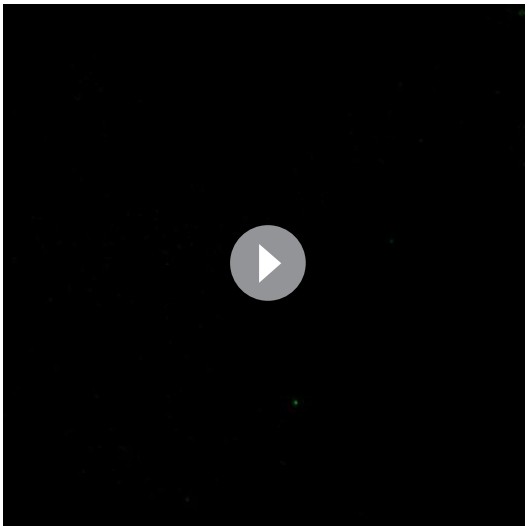

**Video 5.** Calcium imaging of ISCs expressing *trpA1* RNAi in response to AITC. Midguts expressing GCaMP6s reporter together with *trpA1* RNAi in ISCs were dissected and imaged in AHL buffer. Z-stack images were acquired with 10 s interval. A maximal intensity z-projection was shown in the movie. A final concentration of 0.03% TRPA1 agonist AITC was added at 150 s (the 15th frame of the movie).

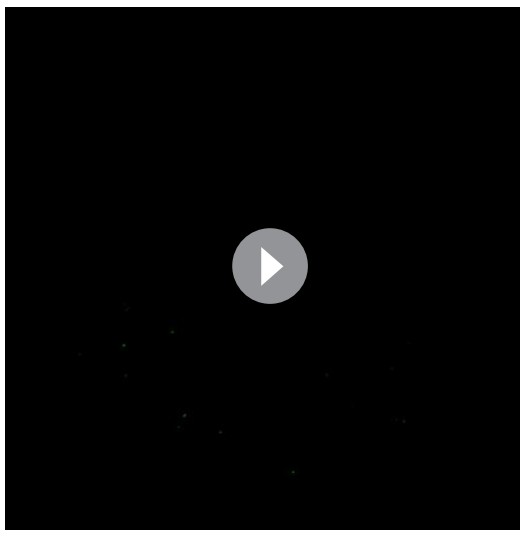

**Video 6.** Calcium imaging of ISCs expressing *RyR* RNAi in response to AITC. Midguts expressing GCaMP6s reporter together with *RyR* RNAi in ISCs were dissected and imaged in AHL buffer. Z-stack images were acquired with 10 s interval. A maximal intensity z-projection was shown in the movie. A final concentration of 0.03% TRPA1 agonist AITC was added at 150 s (the 15th frame of the movie).

the Pvr ligand *pvf1* (**Bond and Foley, 2012**) were suppressed by *trpA1* RNAi and up-regulated in midguts expressing *SERCA* RNAi in ISCs for 1 day to mimic early stages of $Ca^{2+}$ induction (**Figure 7—figure supplement 2**). Altogether, these results suggest that positive RTK signaling feedback loops may play a role in the activation of Ras/MAPK in the context of $Ca^{2+}$ signaling.

## Discussion

### ROS induces ISC proliferation via cation channel-mediated $Ca^{2+}$ signaling

We have found that the two cation channels TRPA1 and RyR are critical for cytosolic $Ca^{2+}$ signaling and ISC proliferation. Under homeostatic conditions, the basal activities of TRPA1 and RyR are required for maintaining cytosolic $Ca^{2+}$ in ISCs to ensure their self-renewal activities and normal tissue turnover (**Figure 8A**). Agonists, including but not limited to low levels of ROS, could be responsible for the basal activities of TRPA1 and RyR. Under tissue damage conditions, increased ROS stimulates the channel activities of TRPA1 to mediate increases in cytosolic $Ca^{2+}$ in ISCs (**Figure 8B**). As for RyR, besides its potential to directly sense ROS, it is known to act synergistically with TRPA1 in a positive feedback mechanism to release more $Ca^{2+}$ from the ER into the cytosol upon sensing the initial $Ca^{2+}$ influx through TRPA1 (**Pan et al., 2016**).

Our findings on the role of $Ca^{2+}$ signaling for ISC proliferation are consistent with a recent study showing that high levels of cytosolic $Ca^{2+}$ correlate with ISC proliferation, and that a prolonged increase in cytosolic $Ca^{2+}$ concentration is sufficient to trigger ISC proliferation (**Deng et al., 2015**). Here, we demonstrate that TRPA1 senses damage-induced ROS; that knockdown of *trpA1* or *RyR* efficiently reduces $Ca^{2+}$ induction in the ISCs caused by ROS; and that cytosolic $Ca^{2+}$ induction is required for damage-induced ISC proliferation.

### Regulation of cytosolic $Ca^{2+}$ in ISCs

Previously, Deng et al. identified L-glutamate as a signal that can activate metabotropic glutamate receptor (mGluR) in ISCs, which in turn modulates the cytosolic $Ca^{2+}$ oscillation pattern via phospholipase C (PLC) and inositol-1,4,5-trisphosphate ($InsP_3$). Interestingly, L-glutamate and *mGluR* RNAi mainly affected the frequency of $Ca^{2+}$ oscillation in ISCs, while their influence on cytosolic $Ca^{2+}$ concentration was very weak. Strikingly, the number of mitotic cells induced by L-glutamate (i.e. an increase from a basal level of ~5 per midgut to ~10 per midgut) is far less than what has been observed in tissue damage conditions (depending on the severity of damage, the number varies from ~20 to more than 100 per midgut following damage) (**Amcheslavsky et al., 2009**; **Chatterjee and Ip, 2009**; **Jiang et al., 2011**; **Karpowicz et al., 2010**). Consistent with this, in our screen for regulators of ISC proliferation in response to tissue damage, we tested three RNAi lines targeting *mGluR* (BL25938, BL32872, and BL41668, which was used by Deng et al.), and none blocked the damage response in ISCs, suggesting that L-glutamate and mGluR do not play a major role in damage repair of the gut epithelium.

We find that ROS can trigger $Ca^{2+}$ increases through the redox- sensitive cation channels TRPA1 and RyR under damage conditions. In particular, we demonstrate using voltage-clamp experiments that the TRPA1-D isoform, which is expressed in the midgut, is sensitive to the oxidant agent paraquat. In addition, the results of previous studies have demonstrated the direct response of RyR to

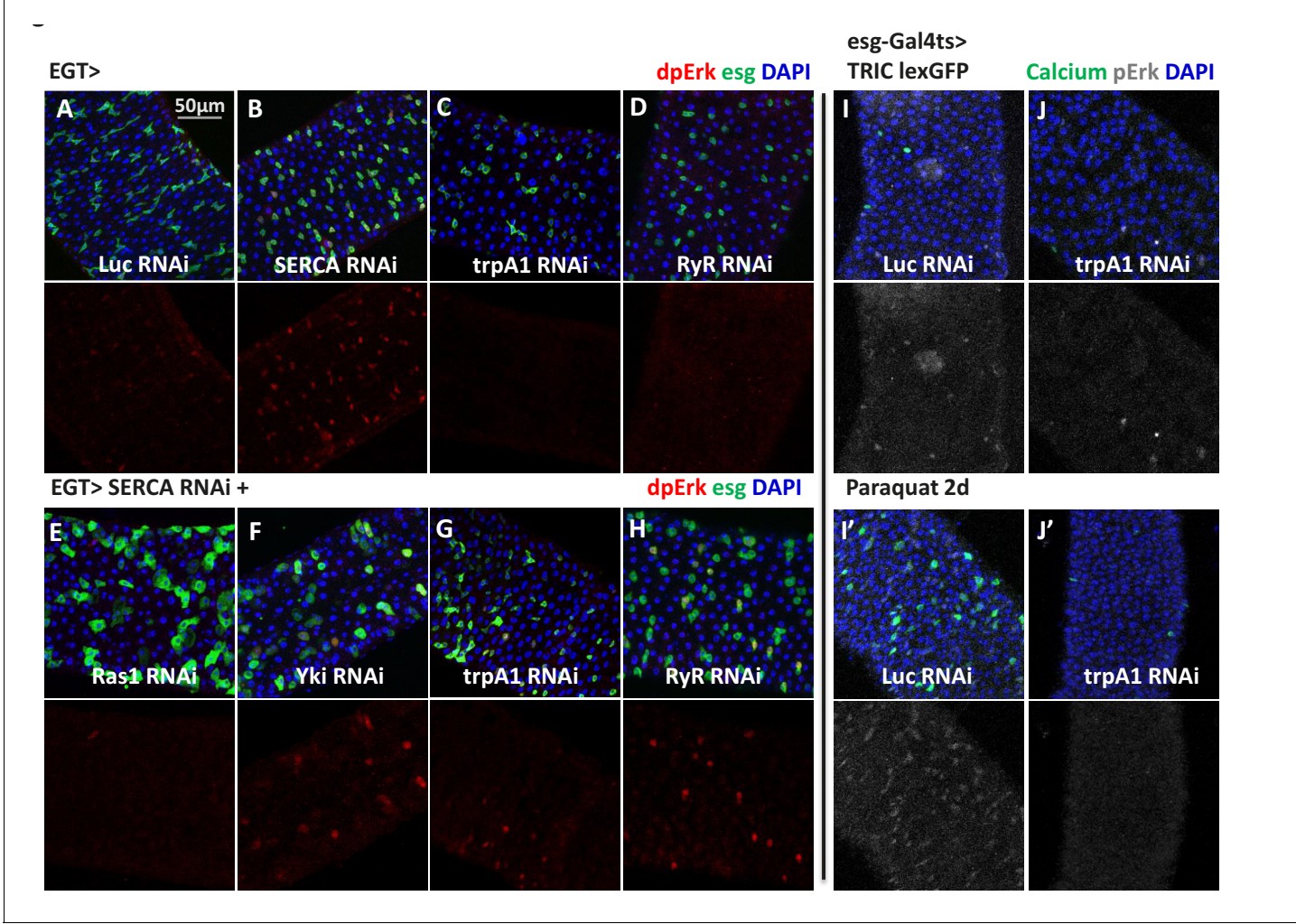

**Figure 6.** High cytosolic Ca$^{2+}$ is necessary and sufficient to activate Ras/MAPK in ISCs. (A–D) Midguts expressing *Luc* RNAi, *SERCA* RNAi, *trpA1* RNAi, or *RyR* RNAi in ISCs for 2-3d are stained for the Ras/MAPK activity marker dpErk. The channel of dpErk signal is shown below the merged image. Note *trpA1* RNAi group was imaged on a different date, but alongside *Luc* RNAi which exhibited the same level of pErk signal as the control group presented in A. (E–H) Midguts expressing *SERCA* RNAi together with *Ras1* RNAi, *Yki* RNAi, *trpA1* RNAi, or *RyR* RNAi in ISCs for 3d are stained for dpErk. The channel of dpErk signal is shown below the merged image. Note *trpA1* RNAi group in G was imaged on a different date together with C. (I–J) Midguts expressing the TRIC lexGFP reporter of intracellular calcium concentration and *Luc* RNAi or *trpA1* RNAi Luc in ISCs for 5d, with the last 2d feeding paraquat (I′–J′), are stained for dpErk. The channel of dpErk signal is shown to the right of the merged image. The TRIC lexGFP reporter consists of lexop-GFP, and two split modules of lexA whose assembly is dependent on intracellular calcium concentration.

The following source data and figure supplements are available for figure 6:

**Source data 1.** Complete results for *Figure 6—figure supplement 2H–I*.

**Figure supplement 1.** Prolonged induction of high cytosolic Ca$^{2+}$ in ISCs results in a nonspecific and variable pattern of Ras/MAPK activation.

**Figure supplement 2.** Src might be a mechanism by which cytosolic Ca$^{2+}$ can activate Ras/MAPK in ISCs.

oxidants via single channel recording (*Terentyev et al., 2008*) and showed that RyR could amplify TRPA1-mediated Ca$^{2+}$ signaling through the Ca$^{2+}$-induced Ca$^{2+}$ release (CICR) mechanism. Interestingly, expression of oxidant- insensitive TRPA1-C isoform in the ISCs also exhibits a tendency to induce ISC proliferation (*Figure 4H*), indicating that ROS may not be the only stimuli for TRPA1 and RyR under physiological conditions. Possible other activators in the midgut may be irritant chemicals,

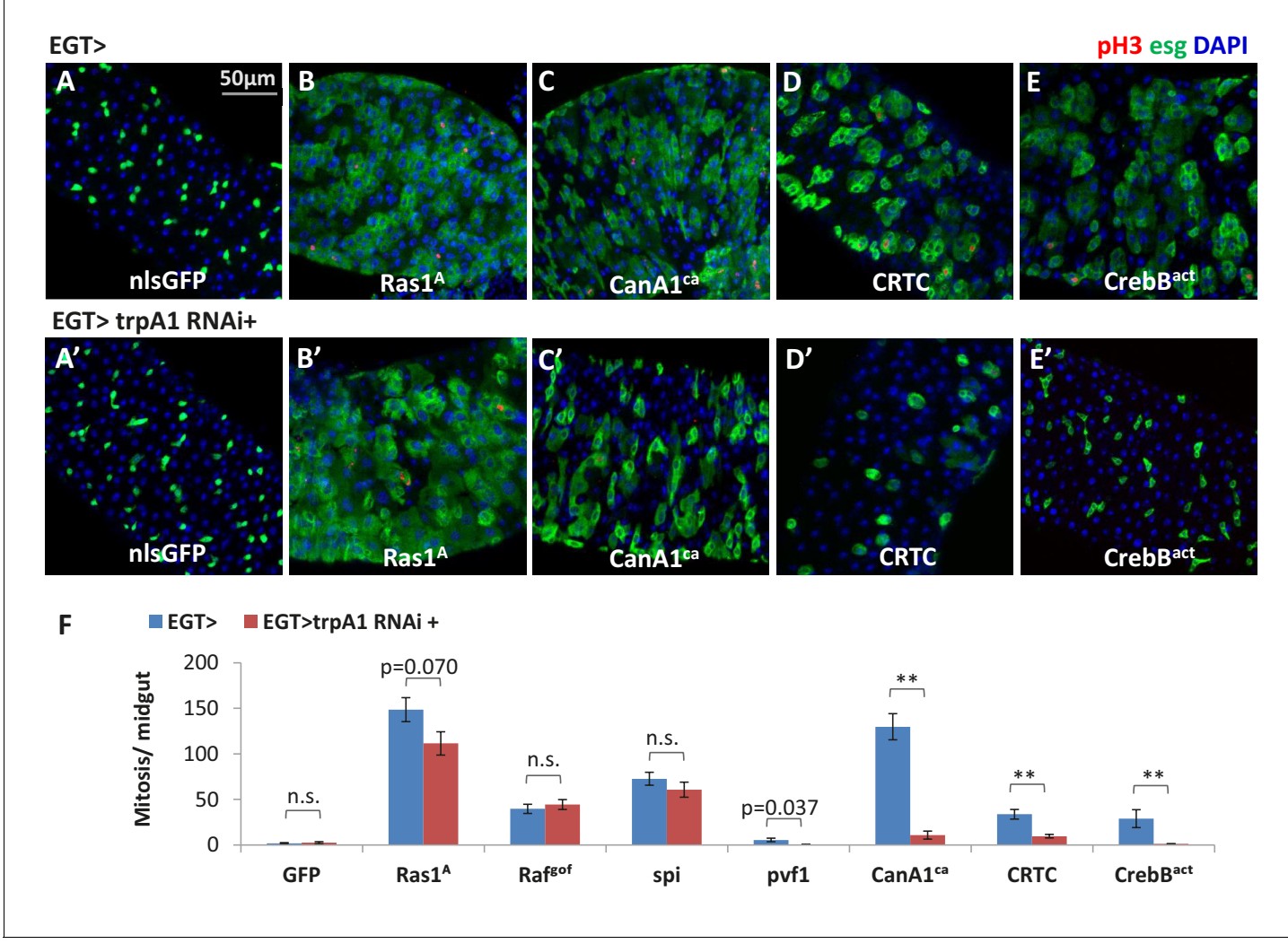

**Figure 7.** Ras/MAPK activity, not CanA/CRTC/CrebB, is sufficient for ISC proliferation in the absence of TRPA1. (A–E) Midguts over-expressing calcium responsive signaling molecules such as active Ras1 (Ras1A), constitutively active Calcineurin A1 (CanA1$^{ca}$), CREB-regulated transcription coactivator (CRTC), or active form of Creb (Creb$^{act}$) in ISCs for 5d are stained for the mitosis marker pH3. The expansion of esg>GFP signal by CanA1$^{ca}$ is reduced by *trpA1* RNAi, although not quite down to the wild-type level. This could be due to the different kinetics of two reagents, as CanA1$^{ca}$ may take effect sooner than *trpA1* RNAi. (A'–E') Midguts expressing *trpA1* RNAi together with calcium-responsive signaling molecules in ISCs for 5d are stained for the mitosis marker pH3. (F) Mitosis quantification of midguts expressing GFP, Ras/ Raf, RTK ligands Spi/ Pvf1, CanA1$^{ca}$, CRTC, Creb$^{act}$, alone or together with *trpA1* RNAi for genetic epistasis analysis. N > 5 midguts are analyzed for each genotype. Data are represented as mean ± SEM. Although it has been reported that *pvf1* overexpression can increase ISC population (*Bond and Foley, 2012*), we could barely detect mitotic effect of Pvf1 in young adult flies.

The following source data and figure supplements are available for figure 7:

**Source data 1.** Complete results for *Figure 7F*, *Figure 7—figure supplement 1F–G*.

**Figure supplement 1.** Ras/MAPK activity, but not CanA1/CrebB, is required for ISC proliferation induced by calcium influx.

**Figure supplement 2.** Ligands for receptor tyrosine kinases (RTKs) are affected by cytosolic Ca$^{2+}$ signaling.

**Figure supplement 3.** Ras/MAPK activity is sufficient for ISC proliferation in the absence of RyR.

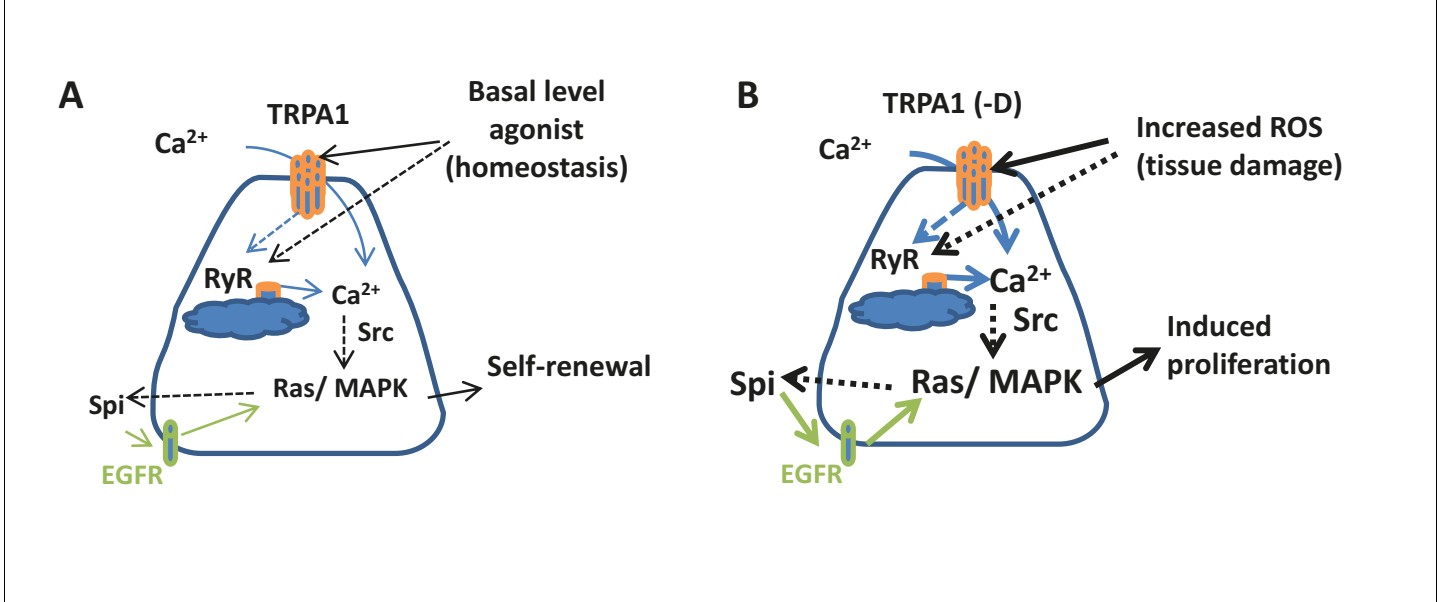

**Figure 8.** Connection between ROS, intracellular calcium, and stem cell activity. (**A**) Under tissue homeostasis conditions, basal levels of ROS and other unidentified stimuli can activate TRPA1 and RyR channels at low levels to allow low levels of cytosolic $Ca^{2+}$, autocrine EGFR ligand Spi, and Ras/MAPK activity required for stem cell self-renewal and minimal level of proliferation. Solid arrows indicate mechanisms that are either well-characterized (green) or demonstrated in this study; dashed arrows indicate unclear mechanisms inferred from literature or this study. (**B**) Under various tissue damage conditions, ROS levels dramatically increase and activate TRPA1, especially the D isoform. RyR channel can be activated either directly by ROS or by the initial $Ca^{2+}$ influx through TRPA1, allowing further calcium release from the ER to the cytosol. High levels of cytosolic $Ca^{2+}$ activate Ras/MAPK signaling via Src, and further amplify Ras/MAPK signaling via autocrine Spi-EGFR signaling. High EGFR-Ras/MAPK activity triggers ISC proliferation.

noxious thermal/mechanical stimuli, or G-protein-coupled receptors (*Fowler and Montell, 2013*; *Shen et al., 2011*; *Zhong et al., 2012*).

Altogether, the concentration of cytosolic $Ca^{2+}$ in ISCs appears to be regulated by a number of mechanisms/inputs including mGluR and the ion channels TRPA1 and RyR. Although mGluR might make a moderate contribution to cytosolic $Ca^{2+}$ in ISCs, TRPA1 and RyR have a much stronger influence on ISC $Ca^{2+}$ levels. Thus, it appears that the extent to which different inputs affect cytosolic $Ca^{2+}$ concentration correlates with the extent of ISC proliferation.

## Signaling events downstream of cytosolic $Ca^{2+}$ in ISCs

Although, as a universal intracellular signal, cytosolic $Ca^{2+}$ controls a plethora of cellular processes, we were able to demonstrate that cytosolic $Ca^{2+}$ levels regulate Ras/MAPK activity in ISCs. Specifically, we found that *trpA1* RNAi or *RyR* RNAi block Ras/MAPK activation in stem cells, and that forced cytosolic $Ca^{2+}$ influx by *SERCA* RNAi induces Ras/MAPK activity. Moreover, Ras/MAPK activation is an early event following increases in cytosolic $Ca^{2+}$, since we observed increased dpErk signal in stem cells expressing *SERCA* RNAi before they undergo massive expansion, and when we co-expressed *Yki* RNAi to block proliferation. It should be noted that a more variable pattern of pErk activation was observed with prolonged increases of cytosolic $Ca^{2+}$, suggesting complicated regulations via negative feedback, cross-activation, and cell communication at late stages of $Ca^{2+}$ signaling. This might explain why Deng et al. failed to detect pErk activation after 4 days induction of $Ca^{2+}$ signaling (*Deng et al., 2015*). Previously, Ras/MAPK activity was reported to increase in ISCs, regulating proliferation rather than differentiation, in regenerating midguts (*Jiang et al., 2011*), which is consistent with our findings about TRPA1 and RyR.

The CanA1/CRTC/CrebB pathway previously proposed to act downstream of mGluR-calcium signaling (*Deng et al., 2015*) is not likely to play a major role in high $Ca^{2+}$-induced ISC proliferation, as we tested multiple RNAi lines targeting *CanA1* or *CrebB* and none of them suppressed *SERCA* RNAi-induced ISC proliferation. In support of this model, we also found that the active forms of CanA1/ CRTC/ CrebB cannot stimulate mitosis in ISCs when their cytosolic $Ca^{2+}$ levels are restricted

by *trpA1* RNAi, whereas mitosis induced by the active forms of Ras or Raf is not suppressed by *trpA1* RNAi.

## Cross-talk between Ca$^{2+}$ signaling and EGFR

Prior to our study, it has been shown that paracrine ligands such as Vn from the visceral muscle, and autocrine ligands such as Spi and Pvf ligands from the stem cells, can stimulate ISC proliferation via RTK-Ras/MAPK signaling (*Bond and Foley, 2012*; *Jiang et al., 2011*; *Xu et al., 2011*). We found that multiple RTK ligands in the midgut are down-regulated by *trpA1* RNAi expression in the ISCs, including *spi* and *pvf1* that can be induced by *SERCA* RNAi. Further, we demonstrated that high Ca$^{2+}$ fails to induce ISC proliferation in the absence of EGFR. As *spi* is induced by EGFR-Ras/MAPK signaling in *Drosophila* cells (*Ammeux et al., 2016*), and DNA binding mapping (DamID) analyses from our lab and others indicate that *spi* might be a direct target of transcriptional factors downstream of EGFR-Ras/MAPK in the ISCs (*Jin et al., 2015*) (Doupé and Perrimon, unpublished), the autocrine ligand Spi might therefore act as a positive feedback mechanism for EGFR-Ras/MAPK signaling in ISCs.

In summary, our study identifies a mechanism by which ISCs sense microenvironment stress signals. The cation channels TRPA1 and RyR detect oxidative stress associated with tissue damage and mediate increases in cytosolic Ca$^{2+}$ in ISCs to amplify and activate EGFR-Ras/MAPK signaling (*Figure 8*). In vertebrates, a number of cation channels (*Monteith et al., 2007*), including TRPA1 and RyR (*Abdul et al., 2008*; *Büch et al., 2013*; *Du et al., 2014*), have been associated with tumor malignancy. Our findings, unraveling the relationship between redox-sensing, cytosolic Ca$^{2+}$, and pro-mitosis Ras/MAPK activity in ISCs, could potentially help understand the roles of cation channels in stem cells and cancers, and inspire novel pharmacological interventions to improve stem cell activity for regeneration purposes and suppress tumorigenic growth of stem cells.

## Materials and methods

### *Drosophila* stocks and culture

The following strains were obtained from the Bloomington *Drosophila* Stock Center: *Dl-lacZ* (BL11651), *y v; attP2* (landing site only, BL36303), *UAS-Luc RNAi* (BL31603), *UAS-trpA1 RNAi* (BL31504, BL31384, BL36780), *UAS-RyR RNAi* (BL29445, BL31540, BL31695), *UAS-SERCA RNAi* (BL25928, BL44581), *trpA1$^1$ (BL26504), UAS-trpA1-A* (BL26263, also called *UAS-TrpA1.K*), *UAS-Ras1 RNAi* (BL29319), *UAS-Yki RNAi* (BL31965), *UAS-Src42A RNAi* (BL44039, validated by *Sopko et al. (2014)*), *UAS-Src64B RNAi* (BL51772), *UAS-EGFR RNAi* (BL25781, BL31525, BL31526), *UAS-CanA1 RNAi* (BL25850, phenotypically validated by *Lee et al. (2016)*), *UAS-CrebB RNAi* (BL63681, BL29332), *UAS-CRTC RNAi* (BL28886), *UAS-rpr* (BL5823), *UAS-Hep$^{ca}$* (BL6406), *UAS-Src42A$^{ca}$* (BL6410), *UAS-Src64B* (BL8477), *UAS-GCaMP6s* (BL42749), *UAS-GCaMP6f* (BL42747), *UAS-RFP Lex-Aop2-GFP; UAS-MKII::nlsLexA$^{DBD}$, UAS-p65AD::CaM; UAS-p65AD::CaM, tubGal80$^{ts}$* (TRIC lexGFP reporter, BL62828), *tubGal80$^{ts}$* (on the X chromosome, BL7016), *tubGal80$^{ts}$[10]* (BL7108), *tubGal80$^{ts}$[20]* (BL7019), *tubGal80$^{ts}$[7]* (BL7018), *UAS-mCherry* (BL52268), *UAS-mCherryCAAX* (BL59021), *RyRGal4$^{R14G09}$* (with cloned enhancer region, BL48662), *FRT19A* (BL1744), *FRT42D; FRT82B* (BL8216), *FRT42D w+* (BL1928), *FRT42D; ry$^{605}$* (BL1802), *FRT2A* (BL1997). Strains from Vienna *Drosophila* RNAi Center: *yw; attP* (landing site only, v60100), *UAS-trpA1 RNAi* (v37249) (*Zhong et al., 2012*), *UAS-Ras1 RNAi* (v106642), *UAS-Yki RNAi* (v104523). Strains from Perrimon lab stock: *w1118, UAS-Luciferase, UAS-nlsGFP, UAS-Ras1$^A$, UAS-Raf$^{gof}$, esgGal4 UAS-GFP tubGal80$^{ts}$* (EGT), *esgGal4, tubGal4, 24BGal4, DaGal4, esgGFP* (generated by David Doupé), *Su(H)GbeGFP* (generated by Li He), *UAS-spi* (secreted form, validated in *Ghiglione et al. (2002)*). *trpA1Gal4$^{CP2A}$* was generated by Junjie Luo. *FRT42D RyR$^{k04913}$* was generated from DGRC111172 (Kyoto Stock Center). *trpA1$^1$ FRT2A* and the stock with a deletion of *trpA1*, Df4415, were obtained from C. Montell lab, *DlGal4* and *Su(H)Gal4* from Steven Hou, *UAS-Cas9.P2* from Fillip Port, *UAS-2xEGFP [m5B29]; UAS-2xEGFP [m6B1] (UAS-ultraGFP)* from Fabio Demontis, *UAS-CncC RNAi* from Dirk Bohmann, *UAS-trpA1-C* (also referred to as *trpA1(A)10b*) and *UAS-trpA1-D* (also referred to as *trpA1(A)10a*) (*Guntur et al., 2015*) from Paul Garrity, *UAS-CanA1 RNAi$^{fb5}$* and *UAS-CanA1$^{ca}$* (constitutively active) (*Wong et al., 2014*) from Patrick H. O'Farrell, *UAS-pvf1* from Edan Foley, *UAS-CRTC* and *UAS-tdTomato-P2A-GCaMP5* from Henri Jasper, *UAS-CrebB$^{act}$* from Jerry Yin. EGT F/O system is

from Bruce Edgar: EGT; UAS-Flp, Act>Stop>Gal4. For MARCM induction, *hsFlp tubGal80 FRT19A/ FM7; tubGal4 UAS-mCD8::GFP/ TM3* was obtained from Ben Ohlstein, *hsFlp tubGal4 UAS-GFP-myc-nls;; tubGal80 FRT2A/ TM6B* from Gary Struhl, and *yw hsFlp UAS-GFP tubGal4; FRT42D tub-Gal80* from Huaqi Jiang.

Flies were reared on standard cornmeal/agar medium in 12: 12 light: dark cycles at 25°C unless noted otherwise. Fly food was changed every 2 days to keep fresh. Conditional expression in adult flies using *tubGal80^{ts}* was achieved by maintaining flies at 18°C until 4–7 days after eclosion, and then shifting young adults to 29°C. For MARCM experiments, flies were maintained at 18°C until 3–5 days after eclosion, heat-shocked at 37°C for 1 hr, and then maintained back at 18°C before dissection and analysis.

See *Supplementary file 1* for detailed genotypes of flies used in each figure.

## Infection and compound feeding

Standard cornmeal/agar medium were melted at 70 ~ 80°C and mixed with a final concentration of 25 µg/ml bleomycin (Calbiochem #203408) or 2 mM paraquat (Sigma Aldrich #856177) to prepare food that can induce tissue damage in the fly midgut. For pathogen infection, *Pseudomonas ento-mophila* (from Tony IP) was grown in LB media at 29°C until reaching $OD_{600}$ = 1, and *Pseudomonas aeruginosa 14* (from Chrysoula Pitsouli) was grown in LB media at 37°C until reaching $OD_{600}$ = 3. Flies were transferred to vials containing a cotton ball impregnated with 5 ml of water solution composed of 20% *Pseudomonas* culture media and 4% sucrose.

## RNAi screen for regulators ISC pool size and damage responses

With a luciferase reporter that we have previously developed as a sensitive surrogate for stem cell pool size (*Markstein et al., 2014*), we performed an RNAi screen to identify novel components regulating ISC activity. In brief, midguts expressing luciferase together with RNAi targeting individual trans-membrane and receptor proteins (*Hu et al., 2015*) in adult ISCs for 9 days, with the last 2 days feeding either normally or on bleomycin food to induce tissue damage. Ten midguts were collected per genotype/ treatment, snap frozen, and stored at −80°C. After samples from all experiment groups were collected, the midguts were homogenized at 4°C using bead blender in Glo Lysis Buffer (Promega) with Halt protease inhibitor (Pierce 87786) and 2 mM trypsin inhibitor benzamidine. The lysate were centrifuged at 8000 rpm for 5 min to sediment the debris. And the supernatants were measured for luciferase activity using the Steady-Glo luciferase assay system (Promega). Luciferase activities were normalized to the value of spiked-in control groups (*yv; attP2* for TRiP stocks, *v60100* for Vienna stocks, and *w1118* for NIG stocks). In particular, we noticed that *trpA1* RNAi (BL31504), *trpA1* RNAi (v37249), and *RyR RNAi* (BL29445) reduced EGT>luciferase activity by 74%, 88%, and 80%, respectively under normal feeding condition, and blocked damage-induced EGT>luciferase expression. We validated the top candidates from the screen by fluorescence imaging.

## Staining and fluorescence imaging

*Drosophila* midguts were dissected in PBS, fixed in 4% paraformaldehyde PBS solution for 30 min, rinsed and kept for 1 hr in PBST solution (PBS with 0.1% BSA, 0.3% Triton X-100) containing 5% Normal Donkey Serum. Midguts were stained overnight at 4°C with the following primary antibodies in PBST solution: rabbit anti-pH3 (Millipore #06–570; 1:3000), mouse anti-GFP (Invitrogen A11120; 1:300), rabbit anti-GFP (Invitrogen A6455; 1:500), rabbit anti-RFP (Life Technologies R10367; 1:500), rabbit anti-*β*-galactosidase (Cappel; 1:6000), rabbit anti-dpErk1/2 (Cell Signaling #4370; 1:500), rabbit anti-Pdm1 (from Xiaohang Yang; 1:500), rat anti-TRPA1 (from Paul Garrity; 1:100). Fly midguts were then washed three times in PBST and incubated with secondary antibodies and DAPI (1:2000) in PBST for 2 hr in the dark at room temperature. Secondary antibodies were donkey anti-rabbit, anti-mouse, and anti-rat IgGs conjugated to Alexa488, Alexa594 and Alexa647 (used at 1:1000, Molecular Probes). Fly guts were then washed three times in PBST, mounted in Vectashield (Vector Laboratories). For anti-TRPA1 staining, primary antibody incubation was done for 1 day and secondary antibody incubation overnight at 4°C. Images of posterior midgut area were captured with a Zeiss LSM780 confocal microscope equipped with 40x oil lens. A z-stack of 10–20 images covering the epithelium from the apical to the basal side were acquired and shown in max projection. All images were adjusted and assembled using NIH ImageJ.

For counting mitotic cells or MARCM clone size, an epifluorescence microscope was used to examine the stained midguts. For the quantification of esg+, Pros+, Dl+ or Su(H)Gbe+ cell density, as well as GCaMP+ cell number, images of the majority of posterior midgut region were acquired on a Keyence microscope.

For measuring midgut length, images of the whole midgut were captured with an epifluorescence microscope. The middle line of the midgut was measured using NIH ImageJ, starting from the end of the proventriculus to the border between the posterior midgut and hindgut, where the malphighian tubules connect to the gut.

For detection of ROS via DHE staining, we followed the previously described protocol (*Hochmuth et al., 2011*). In brief, midguts were dissected and handled throughout in Schneider's medium (HyClone). After incubation in 30 µM DHE (Invitrogen) for 5 min in the dark at room temperature, midguts were washed three times and mounted in Schneider's medium. Images were captured immediately with a Zeiss LSM780 confocal microscope (543 nm excitation, 550–610 nm detection). All images were adjusted and assembled using NIH ImageJ.

## U6-sgRNA for CRISPR/Cas9-mediated conditional knockout

Two pairs of short guide RNAs (sgRNAs) targeting the coding region of *trpA1* or *RyR* were designed using the 'Find CRISPRs' online tool (http://www.flyrnai.org/crispr2/) (*Housden et al., 2016*), and cloned into the double sgRNA vector pCFD4 (*Port et al., 2014*) for site- specific insertion into attP2 on the third chromosome. We obtained transgenic flies ubiquitously (driven by U6 promoters) expressing sgRNAs with the following seed sequences:

U6-sg*trpA1*: CACACGATCCACGTCACTGT & CTGTAGACTCCGTTGTAGAG
U6-sg*RyR*: TTCGTGTTCCATCTGTACAA & ACAAGAACGTGCCGCCGGAT

## RT-qPCR and *trpA1* isoform analysis

Total RNA was extracted from 15 to 20 midguts or 15 heads or 5 larvae (all females) by TRIZOL reagent (Thermo Fisher), converted to cDNA template after DNase I treatment and purification by QIAGEN RNeasy kit. The cDNA was analyzed by real-time quantitative PCR using SYBR Green or by regular PCR (for *trpA1* isoform typing) with *GAPDH* and *rp49* as an internal control. RT-qPCR data were analyzed with Bio-Rad CFX Manager software. Primers used for RT-qPCR or regular PCR (for *trpA1* isoform typing) are shown below:

*trpA1-All*_FW: AGACGCTCATTAAAGTGCTGA
*trpA1-All*_RV: AGAAGGACAAGTGGTTCGGT
*trpA1-CD*_FW: CTTCAGGATATTGCGGGCG
*trpA1-AB*_FW: GTGGACTATCTGGAGGCGG
*trpA1-AB/CD*_RV: CCTTCGCATTAAAGTCGCCA
*trpA1-AD*_FW: TGCCGGCTTTGAATACCATG
*trpA1-BC*_FW: CTTTGGCCGTGGTGAACA
*trpA1-AD/BC*_RV: GCCAGGTGAAAGTACTTGCC
*RyR*_1_FW: GATCGGGACACAGGACATTG
*RyR*_1_RV: GACAGCTTGTCGTTCGACG
*RyR*_2_FW: TGCCTGACCATACCGAGTACA
*RyR*_2_RV: GCCACCAGTCCACTTGGTT
*GAPDH*_FW: CCAATGTCTCCGTTGTGGA
*GAPDH*_RV: TCGGTGTAGCCCAGGATT
*rp49*_FW: ATCGGTTACGGATCGAACAA
*rp49*_RV: GACAATCTCCTTGCGCTTCT
*esg*_FW: ATGAGCCGCAGGATTTGTG
*esg*_RV: CCTCCTCGATGTGTTCATCATCT
*ε-Trypsin*_FW: TAGACACCGGGATACCTCACATCA
*ε-Trypsin*_RV: TCCAGCATTCGCGAGATCCGTATT
*Pdm1*_FW: AGCTGTCCTAACGAGTTCCG
*Pdm1*_RV: ACATCGCGCATATTTGTGTCAA
*pros*_FW: TTTGACCGGAGATGGTGACG
*pros*_RV: GGTCGTTCCTGCCCAGTTTC

spi_FW: CGCCCAAGAATGAAAGAGAG
spi_RV: AGGTATGCTGCTGGTGGAAC
krn_FW: CGTGTTTGGCAACAACAAGT
krn_RV: TGTGGCAATGCAGTTTAAGG
vn_FW: AAGCATCACAAAAGACGTTC
vn_RV: CCGCATCGGAGGAACTATTGA
pvf1_FW: GCGCAGCATCATGAAATCAACCG
pvf1_RV: TGCACGCGGGCATATAGTAGTAG
pvf2_FW: TCAGCGACGAAACGTGCAAGAG
pvf2_RV: TTTGAATGCGGCGTCGTTCC
pvf3_FW: TCGTGAAGAGCAGTAAGCATCG
pvf3_RV: AGGTGCAACTCAGTATGGTGG
dilp3_FW: GCAATGACCAAGAGAACTTTGGA
dilp3_RV: GCAGGGAACGGTCTTCGA
upd1_FW: CAGCGCACGTGAAATAGCAT
upd1_RV: CGAGTCCTGAGGTAAGGGGA
upd2_FW: GACTCTTCTCCGGCAAATCAGA
upd2_RV: TGCTATCGCTGAGGCTCTCG
CRTC_FW: GCAGCAGTACGCAACAAACC
CRTC_RV: TTCACGTCGCTATTGAAGCCC
CanA1_FW: CCGCCAGTGGAAACAAACAG
CanA1_RV: GCGGAAGTGGAACATCATCG

## Two-electrode clamp recordings in *Xenopus* oocytes

TRPA1 currents in *Xenopus laevis* oocytes were recorded as previously described (*Kwon et al., 2010*). *Xenopus* ovaries were dissected and treated with 1.5 mg/ml collagenase A (Roche) and 1 mg/ml trypsin inhibitor (Roche) in OR2 Ringers (100 mM NaCl, 2 mM KCl, 1 mM $MgCl_2$, 5 mM HEPES, pH = 7.5). The deflocculated oocytes were recovered at 18°C for 12 hr in OR3 medium (pH = 7.5): 50% Leibovitz's media L-15 (Sigma L1518), 13 mM HEPES, 90 µg/ml gentamicin, 90 µg/ml Fungizone, 90 µg/ml penicillin/streptomycin. *trpA1-C* and *trpA1-D* (*Kang et al., 2011*) mRNAs were generated from pCS2 or pOX expression vectors with the mMessage mMachine kit (Ambion) and diluted in DEPC-treated water at the concentration of 1 µg/µl. Stage V-VI oocytes were injected with 50 nl prepared mRNA or water, and incubated at 18°C for 3 days for optimal expression. Two electrode voltage clamp recordings were performed at −80 mV with TURBO TEC-03X system and Cellwork software (NPI Electronics, Tamm, Germany), using ND96 buffer (96 mM NaCl, 1 mM $MgCl_2$, 4 mM KCl, and 5 mM HEPES, pH = 7.6) with 1.8 mM $CaCl_2$. In each experiment, we used 4 mM paraquat to treat the oocytes expressing TRPA1-C or -D, and added 0.01% AITC near the end of the recordings as a positive control.

## Imaging cytosolic $Ca^{2+}$ with GCaMP reporters

Cytosolic $Ca^{2+}$ in stem cells was monitored ex vivo using *UAS-GCaMP6f* (*Figure 3E,E', F,F'*), *UAS-GCaMP6s* (*Figure 5* and Supplementary Movies) or *UAS-tdTomato-P2A-GCaMP5G* (*Figure 5—figure supplement 2A,A', B,B'*). In the latter case, tdTomato was used as an internal control. Young adult flies (4–7 days after eclosion) were first incubated at 29°C for 3 days to boost reporter expression before experiments. Intact midguts were dissected and handled in adult hemolymph-like (AHL) buffer. 1–2 intact midguts were placed in single wells of eight-well clear bottom cell culture chamber slides, gently oriented with Nylon mesh, and immersed in 200 µL AHL buffer for imaging. Images of the posterior midgut area (the region close to the middle midgut) were captured with a Zeiss LSM780 confocal microscope equipped with 10x lens. A z-stack of eight images covering one layer of the epithelium was recorded every 10 s for *UAS-GCaMP6s*, or every 20 s for *UAS-tdTomato-P2A-GCaMP5G*. Images before and after drug treatment are shown in z-max projection in *Figure 5—figure supplement 2*, while the time-lapse z-max projection movies for GCaMP6s are shown in *Videos 1–6*. Paraquat (Sigma Aldrich #856177), AITC (Sigma Aldrich #377430), and thapsigargin (Sigma Aldrich T9033) were dissolved in AHL buffer as 5x stock, and 50 µL drug stock was added to 200 µL imaging buffer to obtain desired final concentration. To quantify stem cells with high $Ca^{2+}$ levels in

most of the posterior region, Z-stack images were acquired on a Keyence microscope (*Figure 5*). All images were adjusted, assembled, and analyzed using NIH ImageJ.

## Statistical methods

Statistical analyses were performed using Microsoft Excel (except for qPCR experiments, which were analyzed with Bio-Rad CFX Manager software). Average value and standard error of the mean (SEM) were calculated and shown in the figures. For comparison of data groups, p values were determined by Student's two-tail t-test with unequal variances, except for the comparison before and after drug treatment in *Figure 5D*, where the p value was determined by two-tail paired t-test. Double asterisks (**) label a p value that is less than 0.01. The p value of more than 0.1 is labeled as 'not significant' (n.s.) or not labeled. Sample sizes were chosen empirically based on the observed effects and listed in the figure legends.

## Acknowledgements

We thank the Transgenic RNAi Project (TRiP) and Bloomington Drosophila Stock Center (BDSC) for providing fly stocks; we thank Drs. Paul Garrity, Patrick O'Farrell, Henri Jasper, Jerry Yin, Ben Ohlstein, Gary Struhl, Huaqi Jiang, Dirk Bohmann, and Xiaohang Yang for sharing reagents and fly stocks; we thank the members of the Perrimon lab, especially Kwon Young, Yanhui Hu, David Doupé, Stephanie Mohr, Patrick Jouandin, and Leonard Rabinow, for advice on experiments and comments on the manuscript. Work in the Montell lab is supported by the NIDCD (DC007864) and the NEI (EY010852). Work in the Perrimon lab is supported by NIGMS (GM067761) and HHMI. N.P. is an Investigator of the Howard Hughes Medical Institute.

## Additional information

### Funding

| Funder | Grant reference number | Author |
| --- | --- | --- |
| National Eye Institute | EY010852 | Craig Montell |
| National Institute on Deafness and Other Communication Disorders | DC007864 | Craig Montell |
| Howard Hughes Medical Institute | | Norbert Perrimon |
| National Institute of General Medical Sciences | GM067761 | Norbert Perrimon |

The funders had no role in study design, data collection and interpretation, or the decision to submit the work for publication.

### Author contributions

CX, Conceptualization, Data curation, Formal analysis, Investigation, Visualization, Methodology, Writing—original draft, Writing—review and editing; JL, Investigation, Methodology, Writing—review and editing; LH, Software, Formal analysis, Methodology, Writing—review and editing; CM, Formal analysis, Funding acquisition, Writing—review and editing; NP, Conceptualization, Resources, Supervision, Funding acquisition, Writing—original draft, Project administration, Writing—review and editing

### Author ORCIDs

Norbert Perrimon, http://orcid.org/0000-0001-7542-472X

## Additional files

### Supplementary files

• Supplementary file 1. List of *Drosophila* genotypes used in each figure.

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
