## [Decision Letter]

Thank you for submitting your article "Oxidative stress induces stem cell proliferation via TRPA1/RyR mediated Ca^2+^ signaling in the *Drosophila* midgut" for consideration by *eLife*. Your article has been favorably evaluated by Jonathan Cooper (Senior Editor) and three reviewers, one of whom, Bruce Edgar (Reviewer #1), is a member of our Board of Reviewing Editors. The following individual involved in review of your submission has agreed to reveal their identity: W. Daniel Tracey (Reviewer #2).

The reviewers have discussed the reviews with one another and the Reviewing Editor has drafted this decision to help you prepare a revised submission.

General assessment:

Xu et al. demonstrate that two conserved regulators of intracellular calcium, TrpA1 and RyR, are required for stem cell stress responses in the *Drosophila* gut, and provide evidence that one of these effectors, the Ca^2+^ channel TRPA1, may be activated by reactive oxygen species (ROS). They present an extensive and coherent set of tests supporting a model in which ROS trigger Ca^2+^ release via TRPA1 in ISCs, and that the Ca^2+^ in turn activates Ras/Mapk signaling to promote intestinal stem cell (ISC) divisions. The reviewers agreed that the experiments were generally performed to a high standard and that most of the conclusions were convincingly supported. However, no mechanistic insight was offered as to how ROS might activate TRPA1, or how Ca^2+^ might activate Ras, though similar effects have been reported before in other cell types. Given that other studies of the fly gut have concluded that Ras/Mapk activation in ISCs is effected non-cell autonomously, via the EGFR and ligands produced by other cells, the authors' proposal that Ras is activated cell autonomously by Ca^2+^ is novel, interesting, and provocative.

Essential revisions:

1) Two reviewers noted that authors' proposal that the effect of ROS via Ca^2+^ on ERK and ISC division is cell autonomous, is not well supported and runs counter to many previous publications suggesting that ERK is activated by stress-dependent EGFR ligands non-cell autonomously supplied from other cells in the epithelium. The authors should either provide additional data supporting their cell-autonomous mechanism and ruling out non-autonomous mechanisms, or they should alter the model they present in Figure 7. In any case the autonomous vs. non-autonomous models should be carefully discussed.

2) Along the same lines, reviewer #3 notes that further data should be provided to rule out the role of CrebB and CanA as effectors of Ca^2+^, and to support the authors proposal that ERK is the principal Ca^2+^ effector.

3) Reviewer #2 notes a substantial body of relevant published work that was not cited or discussed (i.e. Du and Guntur papers). The manuscript should be revised to address these deficiencies.

4) Two reviewers commented on the weakness of data showing the TRPA1 activation is sufficient to promote ISC divisions, and reviewer #2 suggests an experiment to test this using a heat-activatable TRPA1 variant. If this experiment is possible, the authors should try it, or present some other type of data further supporting the sufficiency of TRPA1 mediated Ca^2+^ release to promote ISC division. This might also address point #1 above.

5) Reviewers 1and 3 discussed several issues with conclusions that could be clarified by counting ISCs and other cell types (EEs), from experiments that were already performed. This additional analysis should be provided if possible.

*Reviewer #1:*

In this paper Xu et al. demonstrate that two conserved regulators of intracellular calcium, TrpA1 and RyR, are required for stem cell stress responses in the *Drosophila* gut, and provide evidence that one of these effectors, the Ca^2+^ channel TRPA1, may be activated by reactive oxygen species (ROS). They present an extensive and coherent set of tests supporting a model in which ROS trigger Ca^2+^ release via TRPA1 in ISCs, and that the Ca^2+^ in turn activates Ras/Mapk signaling to promote ISC divisions. No mechanistic insight is offered as to how ROS might activate TRPA1, or how Ca^2+^ might activate Ras, though similar effects have been reported before in neurons (also without mechanistic explanation). The results are consistent with a recent report in Nature from Jasper's group (Deng, 2016). Given that other studies of the fly gut have concluded that Ras/Mapk activation in ISCs is effected non-cell autonomously, via the EGFR and ligands produced by other cells, the authors' proposal (Figure 7) that Ras is activated cell autonomously by Ca^2+^ is novel and interesting. However, this study does not definitively rule out non-cell autonomous signaling as a mechanism of ROS sensing or Ca^2+^ release, and so I think the model should be presented with reference to the rather different conclusions of previous work. Otherwise, apart from a few relatively minor issues with data analysis and interpretation, the paper is a nice story in principle appropriate for *eLife*. I list issues that need to be addressed in more detail below:

1) Subsection “TRPA1 and RyR are required for stem cell damage response and self-renewal”, third paragraph, Figure 2: The authors show that the trpA1 and RyR RNAi's deplete Δ+ stem cells in 5 days. Nevertheless, they conclude that these RNAi's are not killing stem cells. This is not very convincing, and I think the number of stem cells should be counted using several assays, at several timepoints, since the issue of cell viability affects other critical results and the overall interpretation. For instance, in Figure 3 we see decreases in ISC mitoses. These results would be confounded if the total number of ISCs decreases during the experiment.

2) Similarly, in Figure 4 we see increases in high Ca^2+^ cells after paraquat treatment in controls, but not in animals expressing RNAi against TRPA1. (Subsection “TRPA1 and RyR are required for normal and ROS-induced cytosolic Ca^2+^ 146 levels in ISCs”, last paragraph, Figure 4—figure supplement 1). But these increases could be due to division of cells with high Ca^2+^, rather than to an increase in the fraction of ISCs with high Ca^2+^. This would not occur in the RNAi treated ISCs, since the RNAi blocks their division. To address this potentially confounding issue the authors should count total ISC numbers in these experiments, as well as numbers of high Ca^2+^ cells, and display the fraction of ISCs that are Ca^2+^ high.

3) As alluded to above, the regulation of Ras activity by Ca^2+^ is not explained mechanistically, and the authors' model that the effect of ROS via Ca^2+^ is autonomous to stem cells is based on correlations that don't rule out other explanations. Although TrpA1 and RyR are clearly necessary for ISC activation, and although other treatments that increase Ca^2+^ are sufficient to activate ISCs (e.g. SERCA-RNAi), this paper doesn't provide strong evidence that activation of TrpA1 or RyR (in this case by ROS) is sufficient to activate ISCs. Moreover the data on ROS activation of TrpA1, though extensive and essential to the model (Figure 7), is the weakest in the paper. Thus, although the authors' model is coherent and consistent with their data, it seems possible that the increase of Ca^2+^ and activation of ERK in ISCs actually involves signals from other cells in the epithelium, rather than a direct effect of ROS on TrpA1, as depicted in Figure 7. It would be interesting, for instance, to test whether the EGFR is required in this context (it is required for ISC responses to many stresses, but should not be required here). In the absence of experiments ruling out roles for other cell types, I feel the model figure (Figure 7) and Discussion should be modified to include the possibility that the oxidative damage signals are received by cells other than stem cells.

4) The dpERK stainings in Figure 5 should be repeated with anti-TrpA1 RNAi's. Only tests with RyR-RNAi are shown.

5) For most of the study, only TrpA1 is tested in detail. The manuscript would be better if the authors had carried through with a deeper analysis of RyR as well

*Reviewer #2:*

The study of Xu et al. finds an important role for the *Drosophila* TRPA1 channel and Ryanodine Receptor channels in Reactive Oxygen Species (ROS) induced proliferation of intestinal stem cells. The combined evidence in support of the role of these two channels in this biological process as presented in the study is very strong. With respect to dTRPA1, the study points to a specific isoform (dTRPA1-D) in the sensing of ROS and the study also finds that Ras/Mapk function downstream of Ca influx/release that are triggered by dTRPA1 and Ryr respectively. The strength of the advance in the study is that it ties calcium signaling to ISC proliferation specifically in the context of ROS triggered proliferation and the study also specifically ties the calcium to dTRPA1 and Ryr. The authors do discuss the fact that a prior study has tied calcium to ISC proliferation through mGluRs and dietary glutamate. But some of the prior literature on dTRPA1 channels (in the context of ROS signaling) has not been adequately credited by the current study. For example, the dTRPA1-D isoform has been previously shown to be activated by ROS (*Drosophila* TRPA1 isoforms detect UV light via photochemical production of H2O2.

Guntur AR, Gu P, Takle K, Chen J, Xiang Y, Yang CH.Proc Natl Acad Sci U S A. 2015 Oct 20;112(42):) through a dual oxidase pathway in the ring gland. As well, a role for dTRPA1 in sensing ROS in the gut (also through dual oxidase) has been previously reported: TrpA1 Regulates Defecation of Food-Borne Pathogens under the Control of the Duox Pathway. Du EJ, Ahn TJ, Kwon I, Lee JH, Park JH, Park SH, Kang TM, Cho H, Kim TJ, Kim HW, Jun Y, Lee HJ, Lee YS, Kwon JY, Kang K.PLoS Genet. 2016 Jan 4;12(1)). This latter study also finds that it is the dTRPA1-D isoform which is activated by ROS and the study has not been cited in the current manuscript. Although Guntur et al. is briefly cited in the current study, the analysis of ROS activation by dTRPA1in heterologous oocytes do not significantly extend those that were initially reported by the studies of Guntur and Du. As such the new heterologous expression analyses should thus be presented as important confirmatory findings (even if investigating different chemical compounds to generate the ROS) with a bit more credit/discussion given to the prior studies. The new biological role for this function- in promoting ISC proliferation in response to ROS- is what is important here, and this will be of great interest to the readers of *eLife*.

Experimental points to be addressed:

The anti-dTRPA1 staining does not provide a clear description of where dTRPA-C/D is actually expressed in the gut. Reporters for dTRPA1-C/D are available from several labs and these could be used to more clearly illustrate the expression pattern.

*Reviewer #3:*

In this manuscript, Xu et al. describe a novel mechanism controlling intestinal stem cell proliferation in *Drosophila*. They found that oxidative stress stimulates Ca^2+^ signaling by regulating TRPA1 and RyR. While the role of calcium signaling has previously been reported, this new study describes a different pathway causing changes in [Ca^2+^] and a different downstream effector cascade (Ras/ERK).

Experiments reported in this paper are generally well designed and properly controlled. They address an important question in the field, regarding the mechanisms that allow the integration of multiple signaling pathways in ISCs to control proliferation rates in response to stress (oxidative stress and tissue damage).

This study has the potential to be of interest for the large audience of *eLife*.

In my opinion, however, a few important points would need to be addressed to fully support the authors' main conclusions:

The authors show that TRPA1 and RyR knock-down results in a loss of Dl+ stem cells. Because they argue that these cells do not die of apoptosis and do not commit to the enteroblast lineage, I am confused about what the fate of these ISCs is, after these genetic manipulations. The authors should at least test whether it promotes enteroendocrine cells fate by monitoring the number of prospero-positive cell in these epithelia. Also, it is possible that the expression of Δ is lost in these stem cells? Other ISC markers such as Sanpodo would be useful to test this hypothesis. Finally, are the single cell trpA and RyR mutant clones Dl-positive?

Altogether, these experiments would greatly help clarifying the fate of ISCs in which trpA or RyR activities is impaired.

The authors use paraquat exposure as an oxidative stress in most of their experiments. While the results are compatible with a regulation of Ca^2+^ signaling by ROS via a cell autonomous mechanism (including the control of TRPA1 activity), relatively simple experiments could be performed to rule out a non-cell autonomous mechanism following paraquat-induced tissue damage.

Does the expression of antioxidant proteins in ISCs prevent ROS-induced Ca^2+^ signaling? Do ISC-specific genetic manipulations that have been shown to induce ROS production in ISCs and promote cell proliferation (such as ND42 RNAi) increase Ca^2+^ signaling?

In the last part of this manuscript, the authors try to exclude the role of CanA1 and CrebB downstream of Ca^2+^ release, arguing that Ras/ERK is the main Ca^2+^ effector in ISCs. This is an important point as it has been previously published that these factors are part of the signaling pathway that regulate ISC proliferation in response to [Ca] changes, and this new study challenges this view. I believe that this conclusion needs to be supported by additional data.

The experiment in Figure 6 needs to be quantified and the other genetic manipulations (multiple RNAi lines targeting CanA1 and CrebB mentioned as "not shown" in the Results and Discussion) should be included in the main figures. It would be essential here to demonstrate that CanA1 and CrebB are dispensable for ROS- and SERCARNAi-mediated proliferation as suggested by the authors' model.

[Editors' note: further revisions were requested prior to acceptance, as described below.]

Thank you for resubmitting your work entitled "Oxidative stress induces stem cell proliferation via TRPA1/RyR mediated Ca^2+^ signaling in the *Drosophila* midgut" for further consideration at *eLife*. Your revised article has been favorably evaluated by Jonathan Cooper (Senior Editor), a Reviewing Editor and two reviewers.

The consensus is that the manuscript has been improved, and we appreciate the new data you've included as well as your editorial efforts. However, there are two remaining issues that are quite significant, and really need to be addressed before acceptance, as outlined below:

1) One reviewer and the reviewing editor still believe the manuscript still falls short of establishing how, mechanistically, [Ca^2+^] activates ERK. A key issue is whether this happens directly, cell-autonomously, or through non-autonomous interactions between different cells in the gut epithelium. The old and new data do not resolve this issue, despite requests from reviewers. The model figure tries to cover both (or all possible) mechanisms and as such is a confusing conclusion – in fact the model as presented will only serve to obscure whatever the real mechanism is, not to resolve it. We would very much like to see a clear resolution of this issue, so that this paper can be seen as an obvious advance over the recent similar publication from Deng et al. A reviewer's specific comments on this point follow: "[…] the authors provide new data demonstrating that [Ca^2+^] controls the expression of Spitz and other RTK ligands. However, the authors appear to favor a model where [Ca^2+^] controls ERK activation directly through Src, but such activation is not demonstrated and the mechanism is unclear. As an example of this, the revised model still highlights a direct regulation of Ras/MAPK by Ca^2+^, and the authors discuss a scenario where Spitz induction is downstream of an initial direct activation of ERK rather than an integral part of the ERK activation mechanism. I think the two models (direct or autocrine/paracrine) could be easily be distinguished by testing whether EGFR and/or PVR are required for ISC proliferation induced by SERCARNAi and TRPA1-D overexpression, as suggested in the initial review. I believe that confirming that Ca^2+^ can activate ERK directly in ISCs (independently of a RTK signal) would greatly increase the impact of this study." Conversely, demonstrating the EGFR is essential for deregulated proliferation after Ca^2+^ signaling dysfunction would at least show that the mechanism is not simple and not cell autonomous.

2) A second, and related, issue is how your results and conclusions relate to the Deng et al. publication. We feel that you should show the relevant data you have and clarify how your conclusions are different from Deng's and why. On this issue, a reviewer commented: "some of the data required is not presented and some of the conflicting data not discussed. For example, the genetic interaction between SERCARNAi and CrebBRNAi (CrebBRNAi not being able to block SERCARNAi-induced proliferation) is still reported as "data not shown" in the revised version. This result is used as an important piece of data to support the authors' conclusion that ERK is the major effector of Ca^2+^ in the control of ISC proliferation. Deng et al. reported that CrebBRNAi expression causes ISC loss: do the authors here find something different? Also, Deng et al. reported that dpERK levels are not affected by SERCARNAi expression. This inconsistency should be at least noted and discussed." Although we feel these issues are less pressing than #1 above, we would appreciate some revisions to make it clear how your results and conclusions differ from those presented by Deng et al.

---

## [Author Response]

*Essential revisions: 1) Two reviewers noted that authors' proposal that the effect of ROS via Ca2+ on ERK and ISC division is cell autonomous, is not well supported and runs counter to many previous publications suggesting that ERK is activated by stress-dependent EGFR ligands non-cell autonomously supplied from other cells in the epithelium. The authors should either provide additional data supporting their cell-autonomous mechanism and ruling out non-autonomous mechanisms, or they should alter the model they present in Figure 7. In any case the autonomous vs. non-autonomous models should be carefully discussed.*

Since it was reported that Ras activation increases Ca^2+^level in the ISCs (Deng et al., 2016), and EGFR ligand spi might act as a positive feedback signal for EGFR/Ras/MAPK pathway (Ammeux et al., 2016), the Ca^2+^mechanism and EGFR mechanism of Erk activation are not mutually exclusive but most likely intertwined. We have revised the model in Figure 7 to reflect the multiple layers of regulation for Ras/MAPK signaling (explained in subsection “Signaling events downstream of cytosolic Ca^2+^in ISCs”, second paragraph).

*2) Along the same lines, reviewer #3 notes that further data should be provided to rule out the role of CrebB and CanA as effectors of Ca^2+^*, *and to support the authors proposal that ERK is the principal Ca^2+^ effector.*

We have quantified the epistasis analysis to demonstrate that CanA1 RNAi cannot reduce ISC proliferation caused by SERCA RNAi (Figure 7—figure supplement 1). In addition, we are presenting the quantification of experiments using the CanA1 RNAi line (“FB5”) from Dr. O’Farrell (Wong et al., 2014). Thus, we are very confident that SERCA RNAi- induced ISC proliferation does not depend on CanA1. In a modified screen to identify suppressor of SERCA RNAi, we have tested RNAi lines against calcium signaling effectors including CrebB (BL29332), CaMKII (BL29401, BL35330) and other calcineurin family proteins like CanB (BL27307), CanB2 (BL27270, BL38971), Pp2B-14D (BL25929), none of which rescued the over-proliferation phenotype.

*3) Reviewer #2 notes a substantial body of relevant published work that was not cited or discussed (i.e. Du and Guntur papers). The manuscript should be revised to address these deficiencies.*

The papers suggested have been cited and discussed in the revised manuscript (see subsection “TRPA1 and RyR expression in the midgut”, first paragraph and subsection “TRPA1-D in the midgut senses the oxidative stress associated with tissue damage”, second paragraph).

*4) Two reviewers commented on the weakness of data showing the TRPA1 activation is sufficient to promote ISC divisions, and reviewer #2 suggests an experiment to test this using a heat-activatable TRPA1 variant. If this experiment is possible, the authors should try it, or present some other type of data further supporting the sufficiency of TRPA1 mediated Ca2+ release to promote ISC division. This might also address point #1 above.*

We have performed the mitosis quantification of midguts overexpressing TRPA1 isoforms C and D, as well as the heat-activatable TRPA1 isoform A (Figure 4). The data support the sufficiency of TRPA1 mediated Ca^2+^release to promote ISC division.

*5) Reviewers 1and 3 discussed several issues with conclusions that could be clarified by counting ISCs and other cell types (EEs), from experiments that were already performed. This additional analysis should be provided if possible.*

We have added the time-course quantification of ISC (Figure 1—figure supplement 1) and EE (Figure 2) cell density in the revised manuscript. Besides, we have used EGT F/O (flp out tracing of ISC progenies) and MARCM systems for lineage tracing, and presented the images of midguts stained for the terminal differentiation markers Pros (for EE) and Pdm1 (for EC) (Figure 2—figure supplement 1). Our data suggest that while TRPA1 and RyR are required for maintaining basal level of cytosolic Ca^2+^and ISC self-renewal, and do not affect ISC differentiation.

*Reviewer #1:*

*[…] 1) Subsection “TRPA1 and RyR are required for stem cell damage response and self-renewal”, third paragraph, Figure 2: The authors show that the trpA1 and RyR RNAi's deplete Δ+ stem cells in 5 days. Nevertheless, they conclude that these RNAi's are not killing stem cells. This is not very convincing, and I think the number of stem cells should be counted using several assays, at several timepoints, since the issue of cell viability affects other critical results and the overall interpretation. For instance, in Figure 3 we see decreases in ISC mitoses. These results would be confounded if the total number of ISCs decreases during the experiment.*

The observation that anti-apoptotic p35 protein cannot rescue ISC proliferation defects (Figure 1—figure supplement 2’-I’), and the added data of anti-cleaved caspase3 stainings (Figure 1—figure supplement 2), demonstrate that trpA1 and RyR RNAi are not causing ISC apoptosis. We have taken the reviewer’s advice and counted ISC and EE density after 3d, 5d, 7d of trpA1 or RyR RNAi expression. Although the signal of Dl-lacZ, indicating ISC maintenance, reduces significantly after 5 days of RNAi expression (Figure 2), the density of overall esg+ stem cell population exhibits a much more moderate decrease (Figure 1—figure supplement 1). Moreover, ISC differentiation activity (measured by Notch reporter Su(H)Gbe-GFP, Figure 2) and EE density (Figure 2) remains relatively stable during the time frame for most of our experiments. Our data suggest that trpA1 or RyR RNAi affect ISC proliferation but not differentiation, therefore the ISC population undergo gradual loss as they differentiate.

*2) Similarly, in Figure 4 we see increases in high Ca^2+^ cells after paraquat treatment in controls, but not in animals expressing RNAi against TRPA1. (Subsection “TRPA1 and RyR are required for normal and ROS-induced cytosolic Ca^2+^ 146 levels in ISCs”, last paragraph, Figure 4—figure supplement 1). But these increases could be due to division of cells with high* Ca^2+^*, rather than to an increase in the fraction of ISCs with high* Ca^2+^*. This would not occur in the RNAi treated ISCs, since the RNAi blocks their division. To address this potentially confounding issue the authors should count total ISC numbers in these experiments, as well as numbers of high* Ca^2+^*cells, and display the fraction of ISCs that are* Ca^2+^*high.*

The increases of high Ca^2+^cells caused by paraquat treatment cannot be caused by expansion of cells with high Ca^2+^because the GCamP imaging lasted less than 20 min, not long enough for cell division. We have also stained flies used in Figure 5 (Figure 4 in original manuscript) with anti-GFP antibody (which recognizes GCamP6s) to visualize total ISC population. We did not observe significant changes in the total ISC number or density by trpA1 RNAi or RyR RNAi expression for the time frame we used to perform the GCamP imaging experiments (Figure 5—figure supplement 1).

*3) As alluded to above, the regulation of Ras activity by Ca2+ is not explained mechanistically, and the authors' model that the effect of ROS via Ca2+ is autonomous to stem cells is based on correlations that don't rule out other explanations. Although TrpA1 and RyR are clearly necessary for ISC activation, and although other treatments that increase Ca2+ are sufficient to activate ISCs (e.g. SERCA-RNAi), this paper doesn't provide strong evidence that activation of TrpA1 or RyR (in this case by ROS) is sufficient to activate ISCs. Moreover the data on ROS activation of TrpA1, though extensive and essential to the model (Figure 7), is the weakest in the paper. Thus, although the authors' model is coherent and consistent with their data, it seems possible that the increase of Ca2+ and activation of ERK in ISCs actually involves signals from other cells in the epithelium, rather than a direct effect of ROS on TrpA1, as depicted in Figure 7. It would be interesting, for instance, to test whether the EGFR is required in this context (it is required for ISC responses to many stresses, but should not be required here). In the absence of experiments ruling out roles for other cell types, I feel the model figure (Figure 7) and Discussion should be modified to include the possibility that the oxidative damage signals are received by cells other than stem cells.*

Mitosis quantification of midguts overexpressing TRPA1 isoforms C and D, as well as the heat-activatable TRPA1 isoform A (Figure 4) demonstrates that TRPA1 gain-of-function can significantly increase ISC division. As the reviewer pointed out, previous published studies have demonstrated that EGFR and its ligands induced after tissue damage can be responsible for Ras/MAPK/Erk activation. We do not think that our mechanism is mutually exclusive with the EGFR-related mechanism. Besides the paracrine ligand vn, EGFR in the progenitor population (ISCs are EBs) can act autonomously by sensing the autocrine ligands spi and Krn (Xu et al., 2011). It was reported before in fly cell lines that spi might act as a positive feedback mechanism for EGFR/Ras/MAPK signaling (Ammeux et al., 2016). Further, on-going work in our lab has identified spi as a target of Ras/MAPK signaling in the ISCs (Doupe et al., unpublished). By RT-qPCR of midgut mRNA we found that the transcription of spi is suppressed by trpA1 RNAi and induced by SERCA RNAi in the ISCs (Figure 6—figure supplement 2). Therefore, TRPA1-mediated *Ca^2+^* might also be required for autocrine activation of EGFR/Ras/MAPK via spi, through or in parallel with the Src-mechanism that we have alluded to. On the other hand, it was reported that Ras activation causes increases in cytosolic *Ca^2+^* level (Deng et al., 2016). We hypothesize that the *Ca^2+^* and EGFR mechanisms are intertwined, amplify each other for activation, and turn off each other if one of them gets restricted. The hypothesis is reflected in the revised model in Figure 8.

*4) The dpERK stainings in Figure 5 should be repeated with anti-TrpA1 RNAi's. Only tests with RyR-RNAi are shown.*

We have added the imaging experiments as suggested (Figure 6). As control, wild type midguts (EGT>Luc RNAi) was processed alongside the experimental groups of Figure 6, and the levels of pErk signal look the same as Figure 6.

*5) For most of the study, only TrpA1 is tested in detail. The manuscript would be better if the authors had carried through with a deeper analysis of RyR as well*

We have added quantification of RyR RNAi effect on EE and EB population (Figure 2); and added the observation that Ras1 is epistatic to RyR RNAi for ISC proliferation (Figure 7—figure supplement 2). Because RyR is not at the plasma membrane, we cannot perform the direct measurement of RyR response to ROS as we did with TRPA1, due to the limitation of our equipment.

*Reviewer #2:*

*The study of Xu et al. finds an important role for the Drosophila TRPA1 channel and Ryanodine Receptor channels in Reactive Oxygen Species (ROS) induced proliferation of intestinal stem cells. The combined evidence in support of the role of these two channels in this biological process as presented in the study is very strong. With respect to dTRPA1, the study points to a specific isoform (dTRPA1-D) in the sensing of ROS and the study also finds that Ras/Mapk function downstream of Ca influx/release that are triggered by dTRPA1 and Ryr respectively. The strength of the advance in the study is that it ties calcium signaling to ISC proliferation specifically in the context of ROS triggered proliferation and the study also specifically ties the calcium to dTRPA1 and Ryr. The authors do discuss the fact that a prior study has tied calcium to ISC proliferation through mGluRs and dietary glutamate. But some of the prior literature on dTRPA1 channels (in the context of ROS signaling) has not been adequately credited by the current study. For example, the dTRPA1-D isoform has been previously shown to be activated by ROS (Drosophila TRPA1 isoforms detect UV light via photochemical production of H2O2.*

*Guntur AR, Gu P, Takle K, Chen J, Xiang Y, Yang CH.Proc Natl Acad Sci U S A. 2015 Oct 20;112(42):) through a dual oxidase pathway in the ring gland. As well, a role for dTRPA1 in sensing ROS in the gut (also through dual oxidase) has been previously reported: TrpA1 Regulates Defecation of Food-Borne Pathogens under the Control of the Duox Pathway. Du EJ, Ahn TJ, Kwon I, Lee JH, Park JH, Park SH, Kang TM, Cho H, Kim TJ, Kim HW, Jun Y, Lee HJ, Lee YS, Kwon JY, Kang K.PLoS Genet. 2016 Jan 4;12(1)). This latter study also finds that it is the dTRPA1-D isoform which is activated by ROS and the study has not been cited in the current manuscript. Although Guntur et al. is briefly cited in the current study, the analysis of ROS activation by dTRPA1in heterologous oocytes do not significantly extend those that were initially reported by the studies of Guntur and Du. As such the new heterologous expression analyses should thus be presented as important confirmatory findings (even if investigating different chemical compounds to generate the ROS) with a bit more credit/discussion given to the prior studies. The new biological role for this function- in promoting ISC proliferation in response to ROS- is what is important here, and this will be of great interest to the readers of eLife.*

We cited the studies of Guntur et al. 2015 in the original manuscript but did not discuss it. Now we have cited and discussed both studies (subsection “TRPA1 and RyR expression in the midgut”, first paragraph and subsection “TRPA1-D in the midgut senses the oxidative stress associated with tissue damage”, second paragraph). Guntur et al. found that TRPA1-C and TRPA1-D isoforms are H2O2-sensitive and that their endogenous expressions in the ring gland, or ectopic expression in other tissues, confer UV light sensitivity via sensing UV light-induced H2O2 production. Although Du et al., 2016 reported that TRPA1-D mediates much stronger NaOCl^-^induced current than TRPA1-C, it seems inconsistent that they could rescue uracil-dependent defecation defects in trpA1 mutant flies with TRPA1-C overexpression but not with TRPA1-D. We found that TRPA1-D, rather than TRPA1-C, exhibits response to paraquat (Figure 4), and that TRPA1-D overexpression in ISCs causes a stronger proliferation phenotype than TRPA1-C (Figure 4).

*Experimental points to be addressed:*

*The anti-dTRPA1 staining does not provide a clear description of where dTRPA-C/D is actually expressed in the gut. Reporters for dTRPA1-C/D are available from several labs and these could be used to more clearly illustrate the expression pattern.*

The signal to noise ratio is not ideal for immunostaining with anti-dTRPA1 antibody in the midgut (Figure 3). Personal communication with Dr. KyeongJin Kang (who used the antibody for the Du et al., 2016 paper) also confirmed that the antibody seems quite dirty. To confirm trpA1 expression, we inserted Gal4 at the C terminal of endogenous dTRPA1 with CRISPR/Cas9 (Figure 3—figure supplement 1). With this Gal4 line we could detect trpA1 expression in some ISCs in the posterior midgut areas that is close to the middle midgut region as well as the posterior end of the midgut, where ISC proliferation is most active (Figure 3). We further demonstrated that for trpA1 mutant flies, there is no calcium influx in ISCs in response to the trpA1 agonist AITC (Figure 3). Moreover, knockdown of trpA1 in Dl+ stem cells with Dl-Gal4 inhibits ISC proliferation (Figure 3—figure supplement 2). These data serve as functional evidence that trpA1 is expressed in the ISCs.

*Reviewer #3:*

*[…] In my opinion, however, a few important points would need to be addressed to fully support the authors' main conclusions:*

*The authors show that TRPA1 and RyR knock-down results in a loss of Dl+ stem cells. Because they argue that these cells do not die of apoptosis and do not commit to the enteroblast lineage, I am confused about what the fate of these ISCs is, after these genetic manipulations. The authors should at least test whether it promotes enteroendocrine cells fate by monitoring the number of prospero-positive cell in these epithelia. Also, it is possible that the expression of Δ is lost in these stem cells? Other ISC markers such as Sanpodo would be useful to test this hypothesis. Finally, are the single cell trpA and RyR mutant clones Dl-positive?*

*Altogether, these experiments would greatly help clarifying the fate of ISCs in which trpA or RyR activities is impaired.*

A more careful quantification of esg+ (Figure 1—figure supplement 1), Su(H)Gbe+ (Figure 2), and Pros+ cell density (Figure 2) has been performed. Besides, lineage-tracing experiments with EGT F/O system, or MARCM, suggest that stem cells deficient for trpA1 or RyR can still differentiate into mature progenies (Figure 2—figure supplement 1). These data support our current hypothesis that trpA1 or RyR knockdown allows for ISC differentiation but blocks ISC self-renewal and proliferation. Therefore, trpA1 or RyR RNAi will cause a gradual loss of esg+ stem cells as they differentiate. During the time frame for most of our experiments, total ISC density (esg+) exhibits very modest change with trpA1 or RyR RNAi expression in ISCs, even though the hallmark of ISC self-renewal (Dl+) and damage-induced ISC proliferation decrease significantly. Since the flies lack ISC self-renewal activity, the total ISC population will gradually decrease as the flies age, and eventually be unable to replenish the mature cell types during tissue turnover.

*The authors use paraquat exposure as an oxidative stress in most of their experiments. While the results are compatible with a regulation of Ca^2+^ signaling by ROS via a cell autonomous mechanism (including the control of TRPA1 activity), relatively simple experiments could be performed to rule out a non-cell autonomous mechanism following paraquat-induced tissue damage.*

*Does the expression of antioxidant proteins in ISCs prevent ROS-induced Ca^2+^ signaling? Do ISC-specific genetic manipulations that have been shown to induce ROS production in ISCs and promote cell proliferation (such as ND42 RNAi) increase Ca^2+^ signaling?*

As previously reported, tissue damage can induce ligands expression from the microenvironment, for example the EGFR ligand vn from the visceral muscle, to stimulate ISC proliferation (Jiang et al., 2011). We do not think the non-cell autonomous mechanisms are mutually exclusive with the Ca^2+^ signaling mechanism we have identified. In the revised manuscript we have incorporated the known mechanism via EGFR into our model (Figure 8). Expression of the antioxidant protein CncC in ISC can reduce its proliferation (Hochmuth et al., 2011). Due to the complexity of the genotype we have not performed GCamP imaging to confirm if ROS-induction of Ca^2+^ signaling is decreased by CncC, or if CncC knockdown can induce Ca^2+^ in ISCs, but we have obtained indirect evidence that CnCC RNAi-induced ISC hyper-proliferation status is dependent on trpA1 (Figure 4).

*In the last part of this manuscript, the authors try to exclude the role of CanA1 and CrebB downstream of Ca^2+^ release, arguing that Ras/ERK is the main Ca^2+^ effector in ISCs. This is an important point as it has been previously published that these factors are part of the signaling pathway that regulate ISC proliferation in response to [Ca] changes, and this new study challenges this view. I believe that this conclusion needs to be supported by additional data.*

*The experiment in Figure 6 needs to be quantified and the other genetic manipulations (multiple RNAi lines targeting CanA1 and CrebB mentioned as "not shown" in the Results and Discussion) should be included in the main figures. It would be essential here to demonstrate that CanA1 and CrebB are dispensable for ROS- and SERCARNAi-mediated proliferation as suggested by the authors' model.*

As the reviewer suggested, we quantified mitosis (Figure 7—figure supplement 1) for genotypes used in Figure 7—figure supplement 1 (Figure 6 in original manuscript). Quantification of mitosis for an additional validated RNAi line against CanA1 (FB5 from Dr. O’Farrell, Figure 7—figure supplement 1) also demonstrates that CanA1 RNAi cannot suppress hyper-proliferation caused by SERCA RNAi.

[Editors' note: further revisions were requested prior to acceptance, as described below.]

*The consensus is that the manuscript has been improved, and we appreciate the new data you've included as well as your editorial efforts. However, there are two remaining issues that are quite significant, and really need to be addressed before acceptance, as outlined below:*

*1) One reviewer and the reviewing editor still believe the manuscript still falls short of establishing how, mechanistically, [Ca++] activates ERK. A key issue is whether this happens directly, cell-autonomously, or through non-autonomous interactions between different cells in the gut epithelium. The old and new data do not resolve this issue, despite requests from reviewers. The model figure tries to cover both (or all possible) mechanisms and as such is a confusing conclusion – in fact the model as presented will only serve to obscure whatever the real mechanism is, not to resolve it. We would very much like to see a clear resolution of this issue, so that this paper can be seen as an obvious advance over the recent similar publication from Deng et al. A reviewer's specific comments on this point follow: "[…] the authors provide new data demonstrating that [Ca^2+^] controls the expression of Spitz and other RTK ligands. However, the authors appear to favor a model where [Ca^2+^] controls ERK activation directly through Src, but such activation is not demonstrated and the mechanism is unclear. As an example of this, the revised model still highlights a direct regulation of Ras/MAPK by Ca^2+^, and the authors discuss a scenario where Spitz induction is downstream of an initial direct activation of ERK rather than an integral part of the ERK activation mechanism. I think the two models (direct or autocrine/paracrine) could be easily be distinguished by testing whether EGFR and/or PVR are required for ISC proliferation induced by SERCARNAi and TRPA1-D overexpression, as suggested in the initial review. I believe that confirming that Ca++ can activate ERK directly in ISCs (independently of a RTK signal) would greatly increase the impact of this study." Conversely, demonstrating the EGFR is essential for deregulated proliferation after Ca++ signaling dysfunction would at least show that the mechanism is not simple and not cell autonomous.*

As suggested by the reviewers, we performed the genetic epistasis experiments and found that *EGFR* RNAi can suppress *SERCA* RNAi-induced ISC proliferation (Figure 7—figure supplement 1). In addition, we have shown that activation of EGFR by *spi* over-expression even allows ISC to bypass the requirement of TRPA1 for proliferation (Figure 7). While there are various RTK ligands that might activate Ras/MAPK, sometimes in a redundant fashion, the intensity of signal they trigger differs by several orders of magnitude. In particular, Spi is known as a strong EGFR ligand compared to weak non-autonomous ligand such as Vn (Austin et al., 2014). And *spi* expression is required in the ISCs for neoplastic proliferation caused by *Notch* RNAi (Patel et al., 2015), suggesting a non-redundant role of autocrine Spi-EGFR signaling in supporting ISC proliferation in some contexts. Ongoing work in our lab suggests that *spi* might be the target of transcription factors downstream of EGFR/Ras/MAPK pathway (DamID analysis by Doupé et al.) in the ISCs. Therefore, Spi could be involved in a positive feedback mechanism to enhance EGFR/Ras/MAPK signaling in the ISCs. We are now proposing that both EGFR and [Ca^2+^] are required for MAPK activation and ISC proliferation, and autocrine Spi-EGFR signaling might participate in amplifying MAPK activity, downstream of [Ca^2+^] (revised model in Figure 8). In the revised model, [Ca^2+^]-sensitive Src could be a possible mechanism for the initial stimulation of Ras/MAPK by [Ca^2+^], as the kinetics of which is considered to be much faster than transcriptional regulation in the neurons (Randlett et al., 2015; Rosen et al., 1994). We have added the mitosis quantification showing that *Src* is epistatic to *trpA1* RNAi (Figure 6—figure supplement 2), and that *Src64B* RNAi can partially suppress *SERCA* RNAi-induced ISC proliferation (Figure 6—figure supplement 2).

Specifically, to address the issue raised by the reviewer we have now revised Figure 6—figure supplement 2, Figure 7, Figure 7—figure supplement 1, Figure 8, reassigned Figure 7—figure supplement 2, adjusted the figure legends accordingly, and added the following paragraphs to the main text:

“In neurons, a common mechanism by which Ca^2+^ regulates Ras/MAPK is through Ca^2+^ sensitive Src kinases (Cullen and Lockyer, 2002), in a process faster than any transcriptional response (Randlett et al., 2015; Rosen et al., 1994). […] These data suggest that Src might mediate the activation of Ras/MAPK by cytosolic Ca^2+^ in ISCs.”

“Previous studies have documented that receptor tyrosine kinase (RTK) EGFR is required for MAPK activity and ISC proliferation (Jiang et al., 2011). […] Altogether, these results suggest that positive RTK signaling feedback loops may play a role in the activation of Ras/MAPK in the context of Ca^2+^ signaling.”

“Cross-talk between Ca^2+^ signaling and EGFR

Prior to our study, it has been shown that paracrine ligands such as Vn from the visceral muscle, and autocrine ligands such as Spi and Pvf ligands from the stem cells, can stimulate ISC proliferation via RTK-Ras/MAPK signaling (Bond and Foley, 2012; Jiang et al., 2011; Xu et al., 2011). […] As *spi* is induced by EGFR-Ras/MAPK signaling in *Drosophila* cells (Ammeux et al., 2016), and DNA binding mapping (DamID) analyses from our lab and others indicate that *spi* might be a direct target of transcriptional factors downstream of EGFR-Ras/MAPK in the ISCs (Jin et al., 2015), the autocrine ligand Spi might act as a positive feedback mechanism for EGFR-Ras/MAPK signaling in ISCs.”

*2) A second, and related, issue is how your results and conclusions relate to the Deng et al. publication. We feel that you should show the relevant data you have and clarify how your conclusions are different from Deng's and why. On this issue, a reviewer commented: "some of the data required is not presented and some of the conflicting data not discussed. For example, the genetic interaction between SERCARNAi and CrebBRNAi (CrebBRNAi not being able to block SERCARNAi-induced proliferation) is still reported as "data not shown" in the revised version. This result is used as an important piece of data to support the authors' conclusion that ERK is the major effector of Ca++ in the control of ISC proliferation. Deng et al. reported that CrebBRNAi expression causes ISC loss: do the authors here find something different? Also, Deng et al. reported that dpERK levels are not affected by SERCARNAi expression. This inconsistency should be at least noted and discussed." Although we feel these issues are less pressing than #1 above, we would appreciate some revisions to make it clear how your results and conclusions differ from those presented by Deng et al.*

As requested by the reviewers, we have included the epistasis analysis of *SERCA* RNAi with *CrebB* RNAi and *CRTC* RNAi (Figure 7—figure supplement 1). In addition, as the reviewers pointed out, Deng et al. claimed that dpERK levels are not affected by *SERCA* RNAi expression (in Extended Figure 9C from their paper, Figure 9). To address the inconsistency, we found that after ISCs start massive expansion with prolonged high Ca^2+^ induction, pErk returns to normal levels in many ISCs (probably through some negative feedback mechanism) and is non-autonomously induced in some ECs (Figure 6—figure supplement 1), resulting in a diffusive and more variable pattern of activation. Such kinetics we observed might explain why Deng et al. failed to detect pErk induction, especially when they were imaging very small fields of 2-3 cells for midgut samples at late stages of Ca^2+^ signaling.

Author response image 1.**DOI:**
http://dx.doi.org/10.7554/eLife.22441.038

Specifically, to address the issue raised by the reviewer we have now added Figure 6—figure supplement 1, revised Figure 7—figure supplement 1, adjusted the figure legends accordingly, and added the following paragraphs to the main text:

“It should be noted that the best timing to detect autonomous activation of dpErk by high Ca^2+^ in the ISCs is before they enter the hyper-proliferative stage. Once ISCs start to expand massively following prolonged high Ca^2+^ induction, pErk returns to normal levels in many ISCs and is non-autonomously induced in some ECs, resulting in a diffusive activation pattern (Figure 6—figure supplement 1).”

“Furthermore, while *Ras1* RNAi could suppress *SERCA* RNAi-induced ISC proliferation, neither *CanA1* RNAi nor *CrebB* RNAi could (Figure 7—figure supplement 1), and *CRTC* RNAi could only confer moderate suppression of ISC hyper-proliferation (Figure 7—figure supplement 1).”